# An electron-blocking interface for garnet-based quasi-solid-state lithium-metal batteries to improve lifespan

Chang Zhang [1,2], Jiameng Yu [1], Yuanyuan Cui [3] ✉, Yinjie Lv [1], Yue Zhang [1], Tianyi Gao [1], Yuxi He [1], Xin Chen [1], Tao Li [4,5], Tianquan Lin [4,5], Qixi Mi [1], Yi Yu [1,2] & Wei Liu [1,2] ✉

Garnet oxide is one of the most promising solid electrolytes for solid-state lithium metal batteries. However, the traditional interface modification layers cannot completely block electron migrating from the current collector to the interior of the solid-state electrolyte, which promotes the penetration of lithium dendrites. In this work, a highly electron-blocking interlayer composed of potassium fluoride (KF) is deposited on garnet oxide $Li_{6.4}La_3Zr_{1.4}Ta_{0.6}O_{12}$ (LLZTO). After reacting with melted lithium metal, KF in-situ transforms to KF/LiF interlayer, which can block the electron leakage and inhibit lithium dendrite growth. The Li symmetric cells using the interlayer show a long cycle life of ~3000 hours at 0.2 mA cm$^{-2}$ and over 350 hours at 0.5 mA cm$^{-2}$ respectively. Moreover, an ionic liquid of LiTFSI in C$_4$mim-TFSI is screened to wet the LLZTO|LiNi$_{0.8}$Co$_{0.1}$Mn$_{0.1}$O$_2$ (NCM) positive electrode interfaces. The Li|KF-LLZTO | NCM cells present a specific capacity of 109.3 mAh g$^{-1}$, long lifespan of 3500 cycles and capacity retention of 72.5% at 25 °C and 2 C (380 mA g$^{-1}$) with an average coulombic efficiency of 99.99%. This work provides a simple and integrated strategy on high-performance quasi-solid-state lithium metal batteries.

Current lithium-ion batteries (LIBs) based on graphite negative electrodes already could not meet the growing energy demand for poor safety and limited energy density[1–5]. Solid state electrolytes (SSEs) strengthen the battery safety essentially by replacing flammable organic liquid electrolytes and polymer separator[6–10]. Lithium metal, the most attractive negative electrode, shows a high theoretical specific capacity of 3860 mAh g$^{-1}$, a low gravimetric density (0.534 g cm$^{-3}$), and the lowest redox potential (−3.040 V vs. the standard hydrogen electrode)[11–14]. Therefore, a solid-state lithium metal battery is considered one of the most promising energy storage devices for high safety and high energy density. Among various types of SSEs, garnet-type Li$_7$La$_3$Zr$_2$O$_{12}$ (LLZO) showed great promise as a near-future

candidate owing to the high ionic conductivity at room temperature, wide electrochemical window, and excellent physicochemical performance (air stability, lithium stability, and mechanical robustness)[15–19].

However, several practical challenges, such as Li dendrite formation and high interface resistance, obstruct further development severely[20–24]. Recently, a growing number of researchers have recognized that the precipitation of lithium metal inside LLZO accelerates the punctures of lithium dendrites. Due to the intrinsic electronic conductivity of LLZO (10$^{-7}$–10$^{-8}$ S cm$^{-1}$), it is nearly inevitable to the short circuits caused by the penetration of lithium metal dendrites during repeated deposition/stripping processes. Han et al.[25] visualized that the high electronic conductivity of LLZO is mostly responsible for

[1]School of Physical Science and Technology, ShanghaiTech University, 201210 Shanghai, China. [2]Shanghai Key Laboratory of High-resolution Electron Microscopy, ShanghaiTech University, 201210 Shanghai, China. [3]School of Materials Science and Engineering, Shanghai University, 200444 Shanghai, China. [4]School of Materials Science and Engineering, Shanghai Jiao Tong University, 200240 Shanghai, China. [5]Zhangjiang Institute for Advanced Study (ZIAS), Shanghai Jiao Tong University, 201210 Shanghai, China. ✉e-mail: cui-yy@shu.edu.cn; liuwei1@shanghaitech.edu.cn

dendrite formation in SSEs by monitoring the dynamic evolution of Li concentration profiles. Similarly, the Miaofang Chi group indicated that $Li^+$ ions are hence prematurely reduced by electrons at grain boundaries, forming local Li filaments and causing a short circuit[26]. Some effective strategies are reported to improve metal lithium wettability, inhibit Li dendrite growth, and reduce interface resistance by interface modification[20–24]. However, the reported interface layers do not satisfy the insulation requirement, so it is difficult to completely block the migration of electrons from the collector to the interior of the solid electrolyte. In addition, LLZO is easily susceptible to $Li_2CO_3$ formation from exposure to water vapor and carbon dioxide (atmospheric environment). The forming $Li_2CO_3$ will cause the lithiophobic interface with poor interface contact and enlarged interface resistance[27–30]. Moreover, the poor interface contact between Li and LLZO caused the high interfacial resistance and heterogeneous current distribution, which were to blame for lithium dendrite formation. Some approaches were reported to alleviate the issues, such as lithium alloy negative electrode, stable Li-ion conducting layer, and coating metal alloys by melt-quenching process[31–33]. These interlayers could reduce the interface resistance and improve the cycle stability. Furthermore, the high impedances at the solid–solid contact interface between LLZO and the positive electrode is another severe issue[34,35]. Generally, a small quantity of liquid electrolyte is added to wet the positive electrode interface and reduce the interface resistance. Nevertheless, the introduction of liquid electrolytes is likely to increase safety risks as well as a series of side reactions[36–38]. Nowadays, Yu et al.[39] developed a solvent-free garnet-based Li metal battery using molten salt (Li,K,Cs)-FSI (eutectic transition temperature at 45 °C) to wet positive electrode interface. Therefore, modifying and optimizing the negative electrode/positive electrode interface simultaneously is crucial for the further application of garnet-based lithium metal solid-state batteries.

Herein, we report an effective strategy to enhance both Li negative electrode and positive electrode interface stability for $Li_{6.4}La_3Zr_{1.4}Ta_{0.6}O_{12}$ (LLZTO) in quasi-solid-state Li-metal batteries. A thin film of KF with a large band gap and high stability was first deposited on the LLZTO surface. A mixed LiF/KF layer then formed because KF partly contacted with melted Li metal, when LLZO solid electrolyte assembled with Li negative electrode. The in-situ formed LiF/KF dense layer covered the LLZTO pellet completely and served as an electron-blocking layer to inhibit electron leakage and reduce the electronic conductivity of LLZTO. Therefore, the LiF/KF interlayer could hinder lithium dendrite growth at the grain boundary and pores, inhibit local current hot spots, and enhance the critical current density. The dendrite growth and current hot spots always caused Li negative electrode and LLZO interface crack, even short circuits during the Li plating/striping cycling process. This phenomenon was also simulated and verified by the finite-element method and density functional theory (DFT) calculation. Besides, LiTFSI in $C_4$mim-TFSI ionic liquid (also known as low-temperature molten salt) was screened to replace liquid electrolytes and avoid side reactions for wetting the positive electrode interface. Consequently, integrating the interface layer of KF/LiF and ionic liquid produced superior rate performance and cycle life for the solvent-free garnet-based lithium-metal batteries.

## Results and discussion

The nonnegligible intrinsic electron conductivity of LLZO SSEs and electron leakage at Li-LLZO interface are considered as the important reasons for the growth of lithium dendrites[23,24]. The schematic illustration of a garnet-base Li-NCM quasi-solid-state lithium metal battery is shown in Fig. S1 (Supplementary Information). Fluorine element (F) has the largest electronegativity of $\chi = 4$, and potassium element (K) shows the lowest electronegativity of $\chi = 0.82$ among common metallic elements. Meanwhile, KF has a band gap of 5.95 eV > 5 eV (Forbidden band, electronic insulator). Theoretically, the ionic compound

potassium fluorine (KF) has strong structural stability and electronic insulation owing to the largest electronegativity difference value of $\Delta\chi = 3.18$ and a wide band gap of 5.95 eV. When the KF layer comes into contact with melted Li metal, KF will partly transform into LiF. And electronegativity difference value of LiF is $\Delta\chi = 3.0$, and the band gap of LiF is 8.72 eV, showing the structural stability and electronic insulation. Hence, the KF/LiF interlayer can serve as an electron-blocking and stable interface layer to uniform Li-ion flux, and inhibit lithium filament growth and penetration through grain boundaries and voids in LLZO as illustrated in Fig. 1a. The uneven local current density at the interface may trigger the preferential growth of lithium metal dendrite at hotspot areas as illustrated in Fig. 1b. Moreover, the large impedance at LLZTO|positive electrode interface was one of another huge issues for full cells. Normally, a small quantity of liquid electrolyte was added to wet the solid–solid contact interface. Whereas the introduction of liquid electrolytes is likely to increase safety risks as well as a series of side reactions. Therefore, various ionic liquids (IL), such as PY14-TFSI, PP13-TFSI, and $C_4$mim-TFSI, were screened to replace liquid electrolytes in our work[21,22]. Finally, 1 M LiTFSI in $C_4$mim-TFSI ionic liquid was adopted to wet LLZTO|positive electrode interface owing to the low interface resistance and excellent electrochemical performance of Li|KF-LLZTO-IL|NCM cells.

The $Li_{6.4}La_3Zr_{1.4}Ta_{0.6}O_{12}$ (LLZTO) powders were synthesized successfully, and LLZTO pellets were sintered according to our previous works[39,40]. The KF thin film (~50 nm thickness) was deposited on the LLZTO pellet surface (noted as LLZTO-KF) by vacuum thermal evaporation (Fig. S2, Supplementary Information). Crystal structure analysis was performed first by X-ray diffraction (XRD). The cubic phase structure of LLZTO powder, LLZTO pellet, and LLZTO-KF pellet was determined. All diffraction peaks in XRD patterns match well with cubic LLZO (PDF#80-0457), and no impurities were observed as shown in Fig. S3. Morphological characterization and element distribution were detected by scanning electron microscope (SEM) coupled with energy dispersive spectrometry (EDS). In order to compare surface properties intuitively, the LLZTO pellet was covered with half by adhesive tape before vacuum thermal evaporation KF layer, then significant surface contrast was seen from the SEM image in Fig. S4a. Combined with the element distribution mapping in Fig. S4b, the F element was clearly observed on the LLZTO-KF side, while the intensity of Zr, Ta, and O elements became weaker than LLZTO side because of the cover of KF layer. Besides, the cross-sectional SEM image and EDS mappings of LLZTO-KF were shown in Fig. 1c, where the smooth surface and KF modifying layer were observed clearly. Moreover, an atomic force microscope (AFM) was employed to measure the surface morphology of LLZTO-KF pellet. Figure S5a is the AFM topography image of LLZTO-KF, and the corresponding 3D topographic image is displayed in Fig. S5b, indicating the dense and uniform layer. Section analysis was given in Fig. S5c, showing the surface roughness of ~25 nm. For further study of the surface chemical composition, X-ray photoelectron spectroscopy (XPS) was carried out to analyze the LLZTO surface with/without the KF layer. As displayed in Fig. S6a–c, the peaks of K and F elements were perceived obviously on the LLZTO-KF surface, and the peak intensity of Li 1s decreased after KF layer deposition. Time of flight secondary ion mass spectrometry (TOF-SIMS) was carried out to analyze the LLZTO-KF surface. As displayed in Fig. 1d, the depth profiling of LLZTO-KF reveals the evolution of several fragments as the sputtering proceeds. $K^-$ and $KF_2^-$ come from the KF layer on the surface of LLZTO-KF, and $TaO_2^-$ and $ZrO_2^-$ signals represent LLZTO ceramic pellets (substrate). Figure 1e shows the 3D view of element distribution, and Fig. 1f is the element distribution map in the 2D view of the $y$–$z$ plane. The $K^-$ and $KF_2^-$ signals are strong in the beginning, and then gradually decline with the sputtering time (etching depth). On the contrary, the $TaO_2^-$ and $ZrO_2^-$ fragment signals are very weak when they appear, increase almost in parallel with the sputtering time, and finally tend to be stable, which means that the LLZTO-KF surface

was covered with a homogeneous KF interlayer, consisting with the results of XPS. Combining the above characterization data, it was proved that the KF layer was successfully deposited on the LLZTO surface.

Next, the reaction between the KF layer and lithium metal was studied scientifically. For our quasi-solid-state lithium metal cells, the lithium negative electrode was prepared by dipping the LLZTO-KF pellet into molten lithium, where lithium metal was tightly bonded to the pellet, and it was difficult to separate them apart to observe the phase evolution of the interlayer material. In order to observe the reaction product of Li and KF, we have designed two kinds of experiments. (1) KF powder reacts with melted Li metal (Sample 1#): put some KF powder into melted Li metal on a stainless steel substrate with

a 300 °C hotplate for full reaction. (2) KF green pellet reacts with melted Li metal (Sample 2#): make a KF green pellet by a uniaxial pressure of 300 MPa, and put it into melted Li metal for 30 min. Then, take out the KF pellet from melted Li (the KF surface was covered by Li metal), and put it on the 300 °C hotplate for a full reaction. Then, XRD and grazing incidence XRD (GI-XRD) were carried out with the two samples. For Sample 1#, strong XRD peaks of Li (PDF#15-0401) and KF (PDF#85-1314) were collected as shown in Fig. S7a (black line). More importantly, the weak XRD peaks of reaction product LiF (PDF#45-1460) were observed successfully. For a clearer detection of the reaction products, GI-XRD was performed with an incidence angle of 1°. And stronger peaks of LiF were observed clearly (Fig. S7b, black line). For Sample 2#, strong and clear peaks of Li, KF, and LiF were

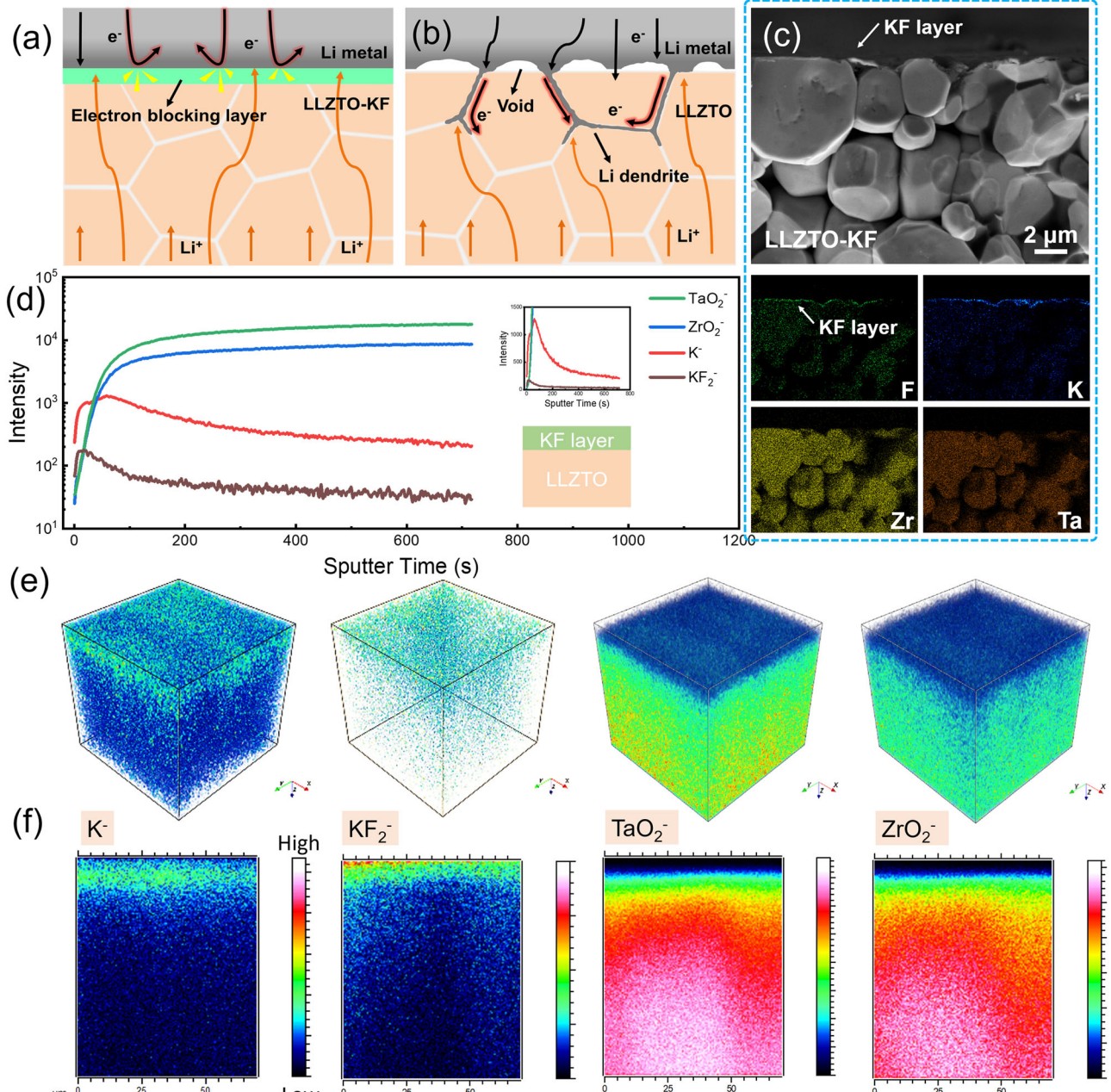

**Fig. 1 | Schematic illustration and interface characterizations of the LLZTO-KF solid-state electrolyte. a**, **b** Schematic illustration of Li|LLZTO-KF interface (**a**) and Li|LLZTO interface (**b**) showing good interfacial contact and no Li dendrite formation due to the electron-blocking buffer layer. **c** Cross-sectional SEM image of LLZTO-KF and corresponding EDS mappings. **d** TOF-SIMS depth profiles for the LLZTO-KF pellet. **e** 3D view of element distribution. **f** 2D view of element distribution of y–z planes.

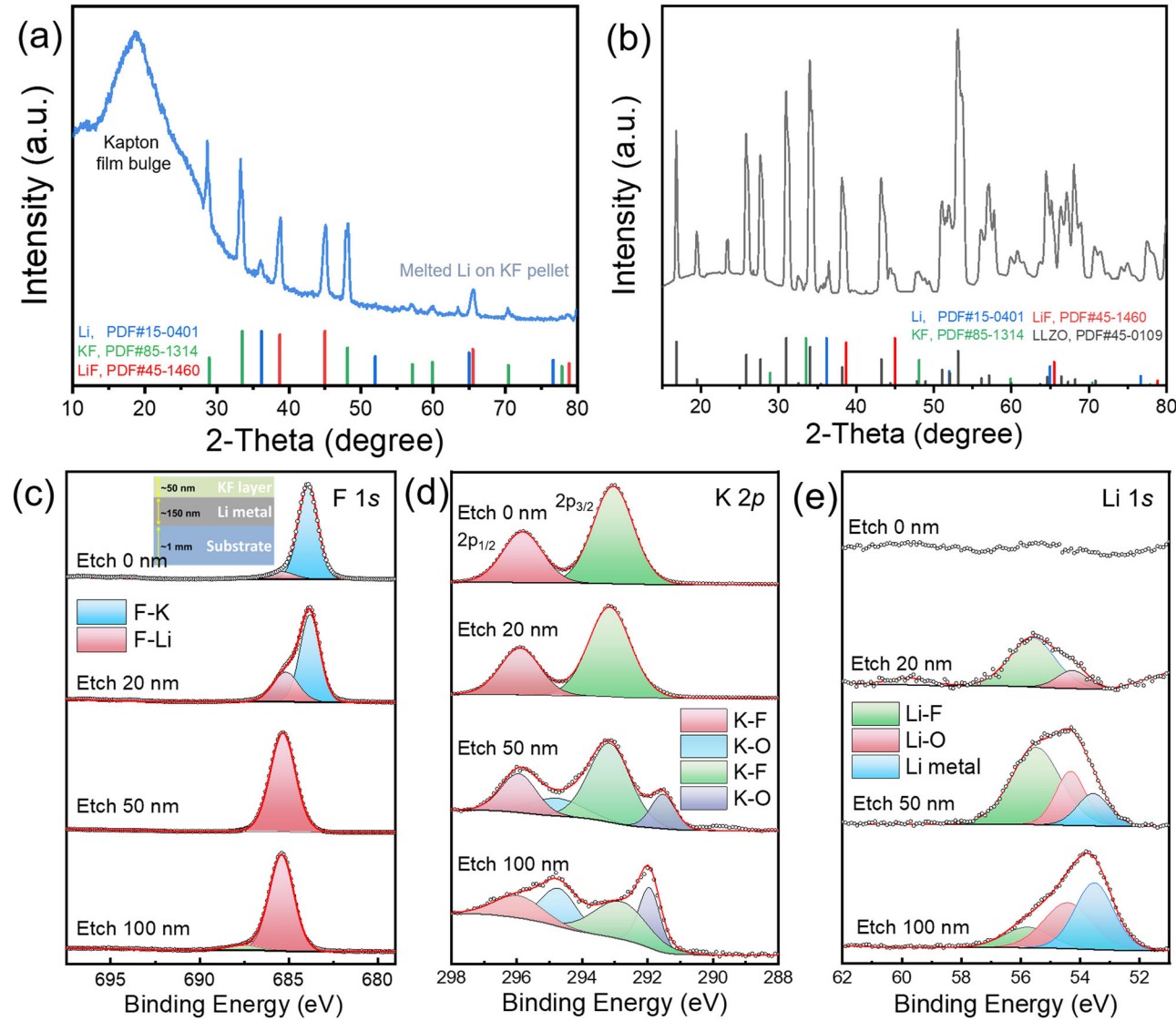

**Fig. 2 | Reaction mechanism of Li and KF buffer layer. a** The GI-XRD patterns with an incidence angle of 1° of Li-KF reaction product under Kapton film. **b** Synchrotron radiation GI-XRD patterns at the incidence angle of 1°. **c–e** XPS spectrum of F 1*s* (**c**), K 2*p* (**d**) and Li 1*s* (**e**) regions with various etching depths of 0, 20, 50, and 100 nm, respectively. The inset in (**c**) is the structure schematic diagram of Substrate|Li|KF samples.

detected from XRD (Fig. S7a blue line) and GI-XRD patterns (Fig. 2a). In addition, the phase structure of Li|KF-LLZTO sample (dipping LLZTO-KF pellet into melted lithium fleetly) was verified by synchrotron radiation GI-XRD with different incidence angle of 0.2°, 0.5° and 1° (Figs. S8 and 2b). Thanks to the high energy and high resolution of the synchrotron radiation X-ray, the phase structure of the interlayer with very low content has been successfully detected. Figure 2b displayed the enlarged synchrotron radiation GI-XRD pattern at 1°, except the LLZTO substrate, the peaks of KF (28.9°, 33.5°, and 48.1°), LiF (38.7° and 45°) and Li (36.2°) were observed clearly.

Moreover, to verify the reaction between the KF layer and Li clearly and easily, depth-profiling X-ray photoelectron spectroscopy (XPS) analysis was performed to obtain the composition information in different depths by ion etching. We first thermally evaporate ~100 nm lithium metal on the substrate and then thermally evaporate a ~50 nm KF layer on the Li metal surface in the glovebox, as shown in Fig. 2c inset. Before etching, as shown in Fig. 2c–e, classical K-F signal was clearly collected on the top layer (KF layer) surface of F 1*s* and K 2*p* regions, while no markedly Li signal was detected on the sample surface. With the etching depth increasing, the signal of generated LiF began to appear and gradually increased according to the XPS

spectrum of F 1*s* and Li 1*s*. Moreover, the K−O signal was collected at 50 nm depth, and it became stronger at 100 nm (Fig. 2d), where the K−O signal comes from the generated K metal (Notably, K is the active metal, always showing K−O signal in XPS result). For Li 1*s* regions, as displayed in Fig. 2e, the lithium metal signal began to appear when the etching depth arrived at the Li metal layer (50 nm), and the lithium metal peak occupied a higher proportion when arriving at the interior lithium layer (100 nm). More intuitive evidence and discussion were given in Fig. S9 and Note S1.

KF can react with molten Li metal at high temperatures according to Eq. (1):

$$Li(l) + KF(s) \rightarrow LiF(s) + K(l) \tag{1}$$

According to the LiF-Li and KF-K phase diagrams, as shown in Fig. S10[41], the forming LiF does not dissolve into Li, and forming K does not dissolve into KF. For our Li|KF−LLZTO interface, during assembling LLZTO-KF with Li metal negative electrode, KF reacted with molten Li metal partially, and the forming LiF could prevent the further reaction between KF and Li at the interface. And, the forming K with very low melting pointing might diffuse into molten Li metal during reaction,

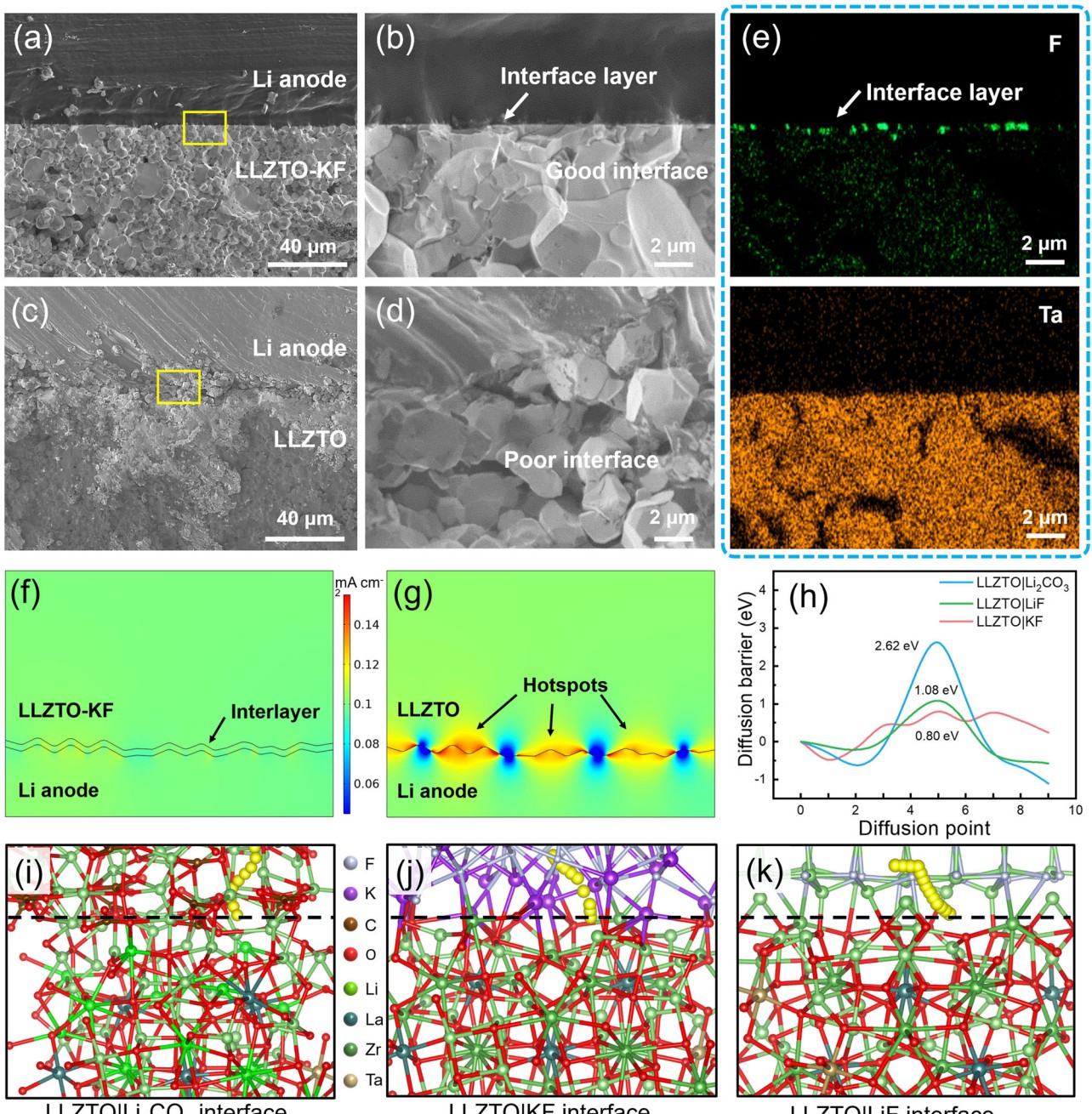

**Fig. 3 | Characterizations and calculations of the interface at Li negative electrode and LLZTO SSEs. a**–**d** Cross-sectional view SEM images of (**a** and **b**) Li|KF−LLZTO interface and (**c** and **d**) Li|LLZTO interface after cycles. Here, the cells were unpacked after CCD tests (0.5 h striping/plating, 0.05 mA cm$^{-2}$/step), and then cross-sectional view SEM was characterized without air exposure. **e** The corresponding EDS mapping images of (**b**). **f** and **g** The finite element analysis results of current density distribution in Li|KF−LLZTO interface (**f**) and Li|LLZTO interface (**g**). The color bar represents the current density. **h** Li diffusion energy with the lowest barrier. **i**–**k** The migration path in LLZTO|Li$_2$CO$_3$ (**i**) LLZTO|KF (**j**), and LLZTO|LiF (**k**) the Li migration path is denoted as yellow balls.

which did not affect the insulation of the interlayer. Therefore, the results of XRD, GI-XRD, and depth-profiling XPS supported the above reaction mechanism. The in-situ reaction of KF and Li metal improves the lithiophilic ability and reduces the interface impedance, resulting in excellent electrochemical performance.

For further prove interface stability, cross-sectional SEM images and EDS mapping were compared between Li|SSE|Li symmetric cells with LLZTO and LLZTO-KF. Before the Li plating and stripping process, both the Li|LLZTO interface and Li|KF−LLZTO interfaces are tight as shown in Fig. S11. After Li plating and stripping cycles, as shown in Fig. 3a, b, a tight and stable interface between lithium negative electrode and LLZTO-KF pellet was detected. By contrast, the lithium metal

and LLZTO cracked severely and formed many voids in the Li|LLZTO interface in Fig. 3c, d, where the voids can cause heterogeneous current density distribution and induce the formation of lithium dendrites. Moreover, the lithium dendrites and deposits were observed at the Li|LLZTO interface (Fig. S12a, b) and inside the LLZTO pellet (Fig. S12c, d) after cycles. When the lithiophilic and electron-blocking interface layer is added, the metallic lithium deposition becomes uniform and flat. In this way, the interface voids and the pulverization of the lithium metal negative electrode are reduced, and the growth of lithium dendrites is restrained. Figure 3e is the corresponding EDS mapping images of the Li|KF-LLZTO interface. F element at the KF/LiF interlayer and Ta element at the LLZTO area were observed visibly,

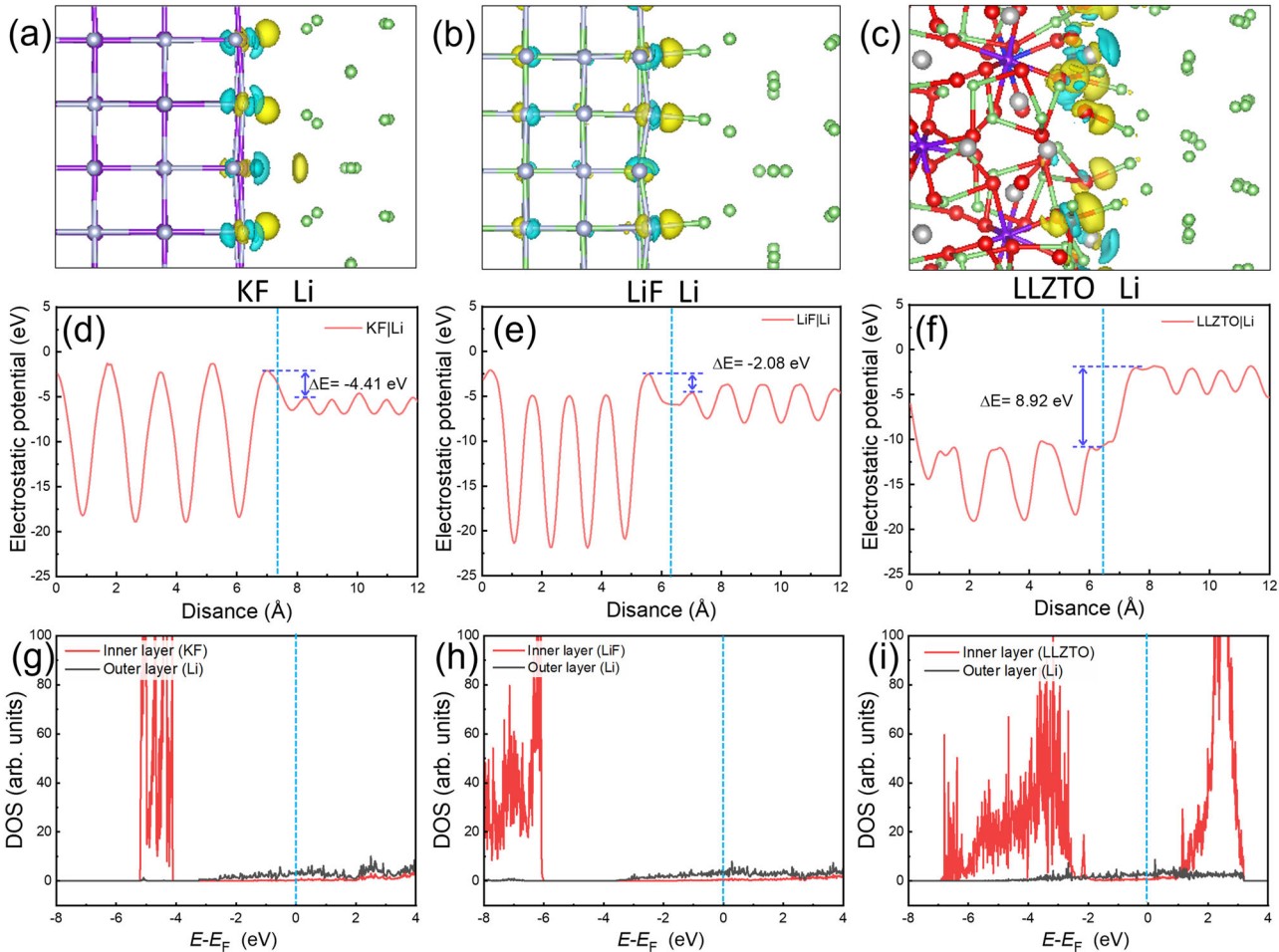

**Fig. 4 | Electron-blocking property of the interface. a–c** The structure and charge transfer of Li|KF interface (**a**), Li|LiF interface (**b**) and Li|LLZTO interface (**c**). The yellow areas indicate electron accumulation, and blue bubbles represent electron depletion, the isosurface value is 0.005 e Å⁻³. **d–f** The corresponding electrostatic potential profiles of the interface. **g–i** The corresponding density of states (DOS) of the interface.

representing good wettability and stability. The stable interface could be attributed to the KF/LiF electron blocking layer, which forbade lithium dendrite growth and reduced local current density at the interface. The finite element analysis based on COMSOL software was employed to further verify the phenomena. As shown in Fig. 3f, the tight Li|KF-LLZTO interface with a modifying layer displayed a homogeneous current density distribution thanks to the KF/LiF interlayer. Correspondingly, the Li|LLZTO interface shows many local current density hotspots (Fig. 3g), in which heterogeneous local current density distribution at the interface may trigger the preferential growth of lithium metal dendrite. Furthermore, DFT calculations were performed to attain deep insight in interface chemistry. The interfacial formation energy of KF|LLZTO was calculated as −2.28 J m⁻² (see the "Methods" section for details), which represents the good interface wettability between LLZTO and KF. Li diffusion energy with the lowest barrier is shown in Fig. 3h. According to a previous report; it is easy for a bare LLZTO surface to form a lithiophobic $Li_2CO_3$ layer with low lithium-ion conductivity in the air[22,27–29]. And $Li_2CO_3$|LLZTO (Fig. 3i) showed a quite high diffusion barrier of 2.62 eV, which may cause lithium accumulation in the local region and then bring about the formation of lithium dendrites. On the contrary, the migration energy barriers of $Li^+$ at KF|LLZTO and LiF|LLZTO (Fig. 3j, k) are calculated as 0.80 and 1.08 eV, respectively, which are lower than that of $Li_2CO_3$|LLZTO (2.62 eV). The low migration energy barriers indicate fast ion transport across the interface, which further facilitates the uniform distribution of $Li^+$ and effectively inhibits the formation of Li dendrites.

In addition, DFT calculations of the Li|KF interface, Li|LiF interface, and Li|LLZTO interface were employed to verify the electron-blocking property. Figure 4a–c are the structure and charge transfer conditions. And the corresponding electrostatic potential profiles of the interface are shown in Fig. 4d–f. KF layer and LiF layer could effectively block electrons at the interface, which is confirmed by the electrostatic potential profiles shown in Fig. 4d, e. The electrostatic potential barriers ($\Delta E$) of −4.41 and −2.08 eV were obtained at the Li|KF interface and Li|LiF interface, respectively. The electrostatic potential barriers indicate the electron-blocking property from Li metal to interlayer. Electrons are contained in the Li metal, and Li deposition occurs preferentially at the interface between lithium and LiF/KF layer, rather than within LLZTO or the surface of LLZTO, which prohibits the penetration of Li dendrites. The electronically insulating nature is further confirmed by DOS results for Li|KF and Li|LiF, shown in Fig. 4g, h and the enlarged figures in Fig. S13a, b in the Supplementary information. In contrast, there is no barrier ($\Delta E$ = 8.92 eV > 0, Fig. 4f) to the transfer of electrons from the Li metal to the bare LLZTO electrolyte. In the case of bare Li|LLZTO interface, electrons and Li atoms preferentially deposit within the LLZTO and this behavior is also corroborated by DOS results (Figs. 4i and S13c).

Next, the electrochemical performances of LLZTO and LLZTO-KF were measured. The EIS of LLZTO and LLZTO-KF samples were measured at various temperatures. Figure S14 exhibited the Nyquist plots of the samples ranging from 0 to 80 °C. Arrhenius plots of Au|LLZTO|Au and Au|KF-LLZTO-KF|Au symmetric cells are shown in Fig. 5a, and

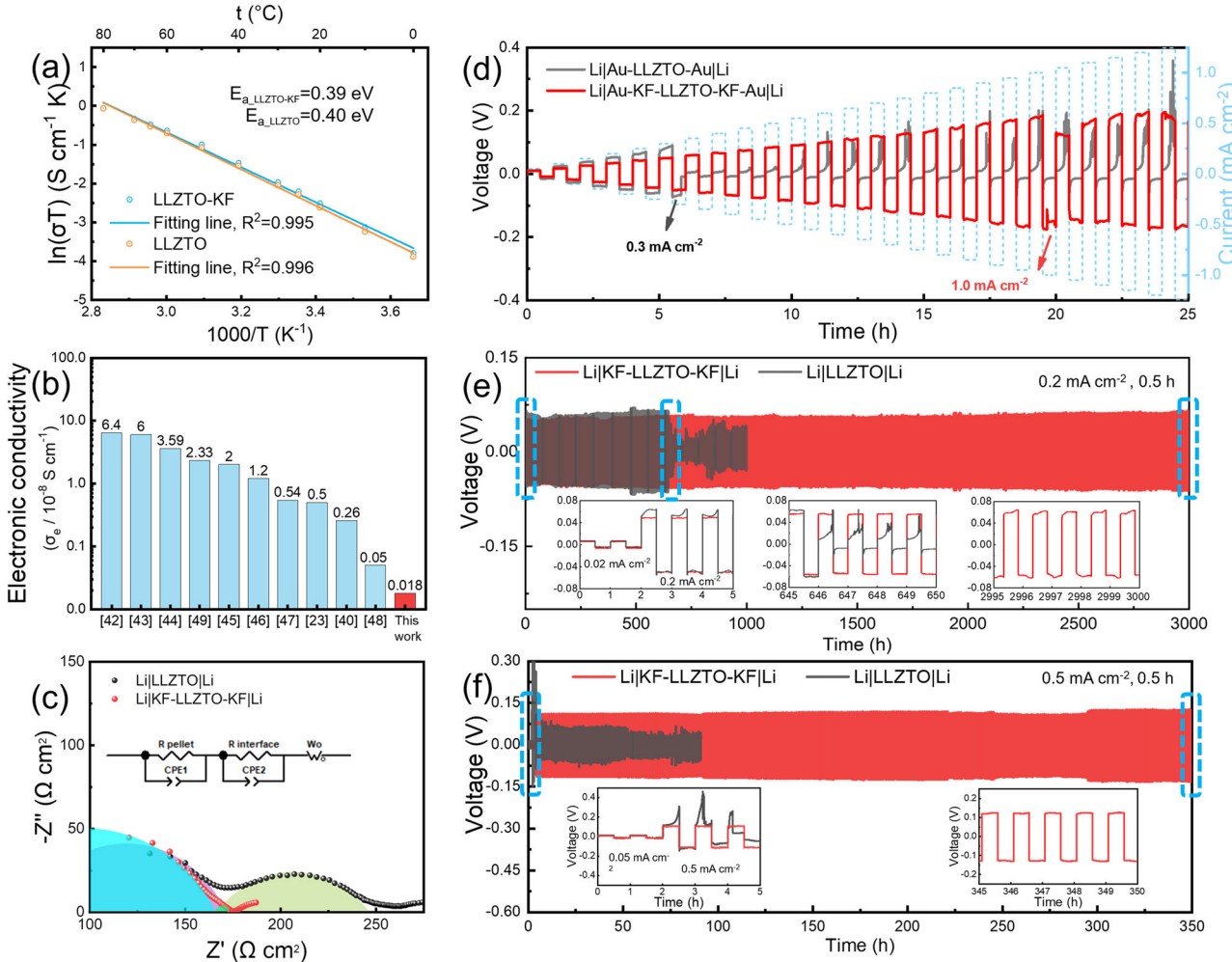

**Fig. 5 | Electrochemical performances of LLZTO solid-state electrolytes and Li symmetric solid-state cells at 25 °C. a** Arrhenius plots of the LLZTO-KF and LLZTO. **b** Electronic conductivity comparison chart of our work with other literature reports. **c** Nyquist plots, the inset is the equivalent circuit. **d** CCD profiles at time-constant mode. **e** Long cycle performance at 0.2 mA cm⁻². **f** Long cycle performance at 0.5 mA cm⁻².

the activation energy of LLZTO and LLZTO-KF is calculated as $0.40 \pm 0.01$ and $0.39 \pm 0.01$ eV, respectively, according to the Eq. (2) (error is estimated in the Supplementary Information Note S2).

$$\sigma = A \exp\left(-\frac{E_a}{kT}\right) \qquad (2)$$

where $\sigma$ is ionic conductivity at different temperatures, $T$ is thermodynamic temperature, $A$ is the pre-exponential factor, $E_a$ is the activation energy, and $k$ is the Boltzmann constant. Electronic conductivity is one of the important factors affecting the nucleation of lithium metal and the generation of lithium dendrites inside the solid-state electrolyte[25,26]. The electronic conductivity was determined by direct current (DC) polarization with a DC voltage of 1 V. As shown in Fig. S15, the steady-state current $I_{ss}$ (contributed by electron transport fully) was reduced from 122.6 to 2.3 nA owing to the stable KF/LiF electron blocking layer. Correspondingly, the electronic conductivity ($\sigma_e$) was calculated by following Eq. (3):

$$\sigma_e = \frac{1}{\rho} = \frac{I_{ss}l}{Us} \qquad (3)$$

where $I_{ss}$ is steady-state current, $l$ is the thickness of LLZTO pellet, $U$ is applying voltage, $s$ is surface area. The electronic conductivity of LLZTO was shortened from $1.4 \times 10^{-8}$ to $1.8 \times 10^{-10}$ S cm⁻¹ after electron-blocking

layer deposition. Figure 5b and Table S1 show the electronic conductivity comparison chart of our work with other literature reports[23,40,42–49], which is the lowest value among reported publications.

Moreover, EIS curves (Fig. 5c) were obtained to assess the interface of Li|LLZTO|Li and Li|KF-LLZTO-KF|Li symmetric cells. The Nyquist plots for the Li symmetric cells with LLZTO show a large semicircle at a medium frequency corresponding to the total charge transfer resistance consisting of two Li|LLZTO interfaces[22]. Different from Li|LLZTO| Li cell, a barely discernible semicircle at medium frequency was observed from Li|KF-LLZTO-KF|Li cells, which meant a low interface resistance and a stable interface contact[50]. The interface resistance of Li|KF-LLZTO-KF|Li symmetric cells was calculated as $5.9\ \Omega\ cm^2$, much lower than $87\ \Omega\ cm^2$ of Li|LLZTO|Li cells. To further evaluate this modified strategy, the critical current density (CCD) test was employed by time-constant mode and capacity-constant mode. Figure 5d was CCD profiles by time-constant mode at 25 °C (0.05 mA cm⁻²/step, 0.5 h plating and 0.5 h striping). Cells with LLZTO-KF exhibited a higher CCD of 1.0 mA cm⁻², while the control sample displayed a low CCD of 0.3 mA cm⁻². Also, the CCD test was performed in capacity-constant mode at 25 °C (0.1 mA cm⁻²/step, 0.2 mAh plating and 0.2 mAh striping). The same increasing tendency occurred toward the CCD tested by capacity constant mode as shown in Fig. S16. The low CCD of 0.6 mA cm⁻² was obtained from cells with LLZTO, while Li|KF-LLZTO-KF|Li cells achieved an increased CCD of as high as 1.4 mA cm⁻². To further verify the stability of the interface layer, the CCD test was also

carried out in time-constant mode at 60 °C (Fig. S17). As expected, cells with LLZTO and LLZTO-KF delivered higher CCD values, where LLZTO showed a CCD of 1.2 mA cm$^{-2}$, and LLZTO-KF displayed a remarkably high CCD value of 1.7 mA cm$^{-2}$. The low CCD of LLZTO could be attributed to the inhomogeneous interfacial contact between LLZTO and Li negative electrode, leading to the uneven local current density (hotspots) and fast dendrite penetration at the interface. The high CCD for modified LLZTO-KF was ascribed to limited electron leakage and uniform local current density according to the extreme electron-blocking KF/LiF interface layer. The experimental results were in good agreement with the COMSOL simulation results (Fig. 3f, g). Furthermore, a long cycle test was carried out to assess the interface stability. Figure 5e shows the long cycle test at the current density of 0.2 mA cm$^{-2}$. Li|LLZTO|Li cells were short-circuited after ~647 h. Satisfyingly, cells with LLZTO-KF worked stably for over 3000 h without any breaks. As for a higher current density of 0.5 mA cm$^{-2}$, Li|KF-LLZTO-KF| Li cells displayed over 350 h of lifespan. In contrast, cells using bare LLZTO were broken after 1 cycle at 0.5 mA cm$^{-2}$ immediately (Fig. 5f). In brief, Li symmetric cells with LLZTO-KF exhibited much longer cycling life, higher critical current density, and lower interface resistance than those of the reported works given in Fig. S18[21–23,40,50–61] and Table S2.

To assess the feasibility and stability of LLZTO-KF solid-state electrolyte in practical application, the quasi-solid-state batteries with NCM-positive electrode, LLZTO or LLZTO-KF SSE, and Li metal negative electrode were assembled, as schematically shown in Fig. S1. For wetting positive electrode interface, various ionic liquids (ILs) of 3 μL, such as PY14-TFSI, PP13-TFSI, and C$_4$mim-TFSI, were screened to replace liquid electrolytes[21,22]. Firstly, the EIS curves of Li|KF-LLZTO-IL| NCM cells with low-concentration ionic liquids were displayed in Fig. S19, 0.2 M LiTFSI in C$_4$mim-TFSI (aliased as 0.2 M C$_4$mim) showed lower interface resistance than another two ILs. As EIS curves shown in Fig. S20, the ILs with high concentration (2 M) could further reduce interface impedance. The smallest interface impedance value of ~76 Ω cm$^2$ (Fig. 6a) was obtained in cells with 1 M C$_4$mim, which is better than cells with 1 M PP13 of 143 Ω cm$^2$ and 1 M liquid electrolyte (LE) of 340 Ω cm$^2$ according to the Figs. S21 and S22. The corresponding interface impedance values are listed in Table S3.

Moreover, the electrochemical performance of garnet-based quasi-solid-state Li|NCM cells with 1 M C$_4$mim was measured. Charge−discharge curves of Li|KF-LLZTO-IL|NCM cells at various C-rate were displayed in Fig. 6b. With the increasing of C-rate, the charging−discharging curves maintain good shapes ranging from 2.8 to 4.3 V with low polarizations at 0.05, 0.1, 0.2, 0.5 and 1 C (1 C = 190 mA g$^{-1}$). The rate test result was exhibited in Fig. 6c (mass loading of NCM is -2.5 mg cm$^{-2}$, here 1 C = 190 mA g$^{-1}$ × 2.5 mg cm$^{-2}$ = 0.475 mA cm$^{-2}$). At the C-rate of 0.05 C, an activation process occurred in the first five cycles, presenting the increasing specific capacity from 183.4 to 192.4 mAh g$^{-1}$. And the specific capacities of 186.8, 177.6, 147.8, 117.5 and 63.3 mAh g$^{-1}$ were realized at 0.1, 0.2, 0.5, 1, and 2 C, respectively. When returned to 0.2 C, a high specific capacity of 176.2 mAh g$^{-1}$ was observed, 96.1% of that at 0.05 C. For the cells with bare LLZTO, the high specific capacity of 193.2 and 179.1 mAh g$^{-1}$ were realized at 0.05 and 0.1 C. However, the capacity decayed rapidly after 0.5 C and was short-circuit at 2 C. As for the long cycle test, for Li|NCM cells with LLZTO-KF, the high specific capacity of 178.2 mAh g$^{-1}$ was realized, with a high capacity retention of 82.0% at 0.2 C after 300 cycles as shown in Fig. 6d. As a contrast, Li| LLZTO-IL|NCM cells show an initial specific capacity of 139.4 mAh g$^{-1}$ at 0.2 C and decay to 117.6 mAh g$^{-1}$ after 221 cycles (Fig. S23). Also, high mass loading NCM of ~4 mg cm$^{-2}$ was adopted and cycled at 0.2 C. As displayed in Fig. S24, the cells with LLZTO-KF have a high initial specific capacity of 145.9 mAh g$^{-1}$, with a capacity retention of 83.8% after 125 cycles.

In consideration of the CCD of LLZTO SSEs and cell polarizations at high current density, the long cycle tests at 1 and 2 C were performed with NCM positive electrode (mass loading is -1 mg cm$^{-2}$).

Cells with LLZTO-KF displayed stable cycle life over 1000 cycles at 1 C, showing an initial specific capacity of 123.1 mAh g$^{-1}$ and capacity retention of 83.1% (Fig. S25). Besides, the excellent capacity retention of 72.5% was realized after 3500 cycles at 2 C and 25 °C (Fig. 6e), representing an average coulombic efficiency of 99.99%. In contrast, cells with bare LLZTO showed a specific capacity of 22.9 mAh g$^{-1}$, and decay to lower than 10 mAh g$^{-1}$ after 45 cycles. To further evaluate cycle stability and electrochemical performance, the cells were tested at 2 C in a 60 °C thermostat. As shown in Fig. S26, a high initial specific capacity of 186.1 mAh g$^{-1}$ was presented in Li|KF-LLZTO-IL|NCM cells, delivering a high capacity of 139.9 mAh g$^{-1}$ after 1000 cycles. While, Li| LLZTO-IL|NCM cells without modifying layer displayed a low initial capacity of only 121.8 mAh g$^{-1}$. Figure S27 and Table S4 give the long-cycle performance comparison results of our work with other literature reports[22,23,57–67]. Both the satisfactory rate performance and long cycle performance were attributed to the extreme electron blocking KF/LiF interlayer and LLZTO|positive electrode interface, demonstrating the stability and feasibility for the implementation of high-performance solvent-free garnet base quasi-solid-state lithium batteries.

In summary, we proposed the integrated strategy that deposits a highly insulating KF layer at Li metal negative electrode interface and 1 M LiTFSI in C$_4$mim-TFSI ionic liquid at positive electrode interface for high-performance lithium-metal batteries using LLZTO solid-state electrolyte. When contacted with melted lithium, KF in situ transforms to a KF/LiF mixture interlayer, which can reduce electronic conductivity, inhibit electron leakage, and then restrain lithium dendrite penetration in LLZTO. As a result, LLZTO-KF exhibited a low electronic conductivity of 1.8 × 10$^{-10}$ S cm$^{-1}$ and a low Li|LLZTO interface resistance of 5.9 Ω cm$^2$. The symmetric cells with LLZTO-KF showed high CCD values of 1.0 or 1.4 mA cm$^{-2}$ by time-constant mode or capacity-constant mode at 25 °C, respectively. And long cycle life was archived with >3000 h at 0.2 mA cm$^{-2}$ and over 350 h at 0.5 mA cm$^{-2}$, respectively. Furthermore, the solid-state cells delivered a high specific capacity of 192.4 mAh g$^{-1}$ at 0.05 C and 25 °C. Especially, the long cycle life of 3500 cycles was realized at 2 C, presenting an average coulombic efficiency of 99.99%. The integration of the KF/LiF layer and C$_4$mim-TFSI ionic liquid was an effective method for garnet-based lithium-metal batteries with good rate performance and long cycle life.

## Methods

Preparation of Li$_{6.4}$La$_3$Zr$_{1.4}$Ta$_{0.6}$O$_{12}$. The Li$_{6.4}$La$_3$Zr$_{1.4}$Ta$_{0.6}$O$_{12}$ (LLZTO) powders were produced by the classical solid-state reaction method according to our previous works[39,40]. In brief, LiOH·H$_2$O (Aladdin), nano La$_2$O$_3$ (Macklin), ZrO$_2$ (Energy Chemical), and Ta$_2$O$_5$ (Aladdin) were dispersed into isopropyl alcohol by stoichiometric ratio and ball milled at 400 rpm for 10 h, in which 20% excess of LiOH·H$_2$O was used to compensate the Li loss during the subsequent calcination process. After ball-milling and drying, the mixture was ground roughly and calcinated at 950 °C for 6 h with a heating rate of 5 °C min$^{-1}$.

### Preparation of LLZTO and LLZTO-KF pellet

The synthetic LLZTO powder and 2 wt% nano γ-Al$_2$O$_3$ (Macklin) were fully ground by a mortar and pestle, where Al$_2$O$_3$ was used as a sintering additive for dense LLZTO pellet. Then -0.5 g fine powders were transferred into a stainless-steel die ($d$ = 12.7 mm) and pressed under a uniaxial pressure of 300 MPa for 2 min. These green pellets were sintered in a MgO crucible at 1250 °C for 40 min in the air with a heating rate of 5 °C min$^{-1}$. As per our previous report, when the LLZTO pellet was stored in Air, Li$_2$CO$_3$ was easy to form on the surface[30]. The LLZTO pellets were grinded and polished with sandpaper and abrasive paste, and ethanol was used to clean these pellets by ultrasonic wave. Then, the fresh pellets were transferred to the glovebox with Ar immediately. Therefore, there is nearly no Li$_2$CO$_3$ contaminant on the surface of our

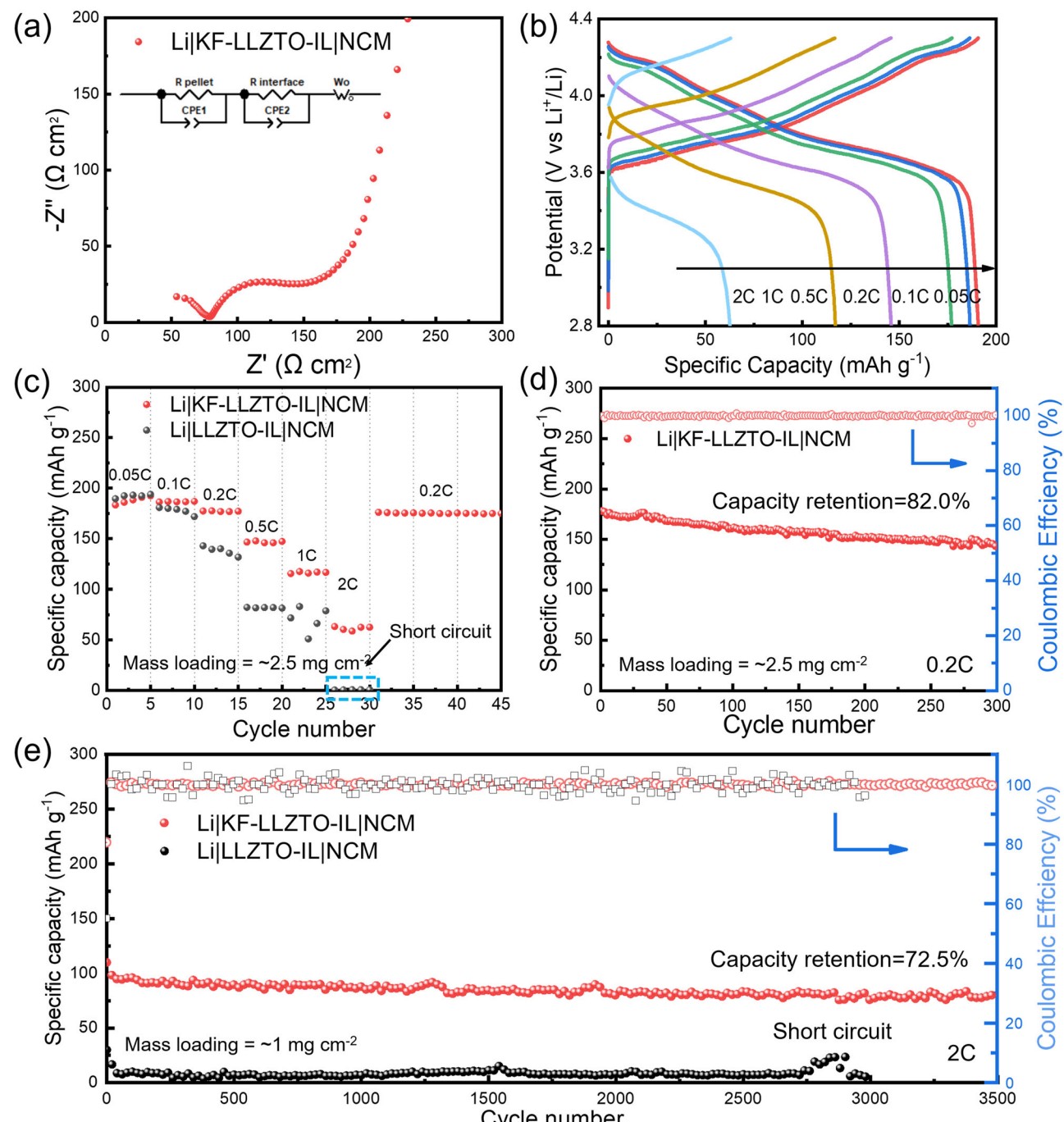

**Fig. 6 | Electrochemical performances of Li|LLZTO-IL|NCM and Li|KF-LLZTO-IL| NCM cells at 25 °C, ranged 2.8–4.3 V. The ionic liquid is 1 M LiTFSI in C₄mim-TFSI. a** Nyquist plot of cells of Li|KF-LLZTO-IL|NCM, the inset is the equivalent circuit. **b** Charge−discharge curves of Li|KF-LLZTO-IL| NCM at various C-rate. **c** Rate capability. **d** Long cycle performance at 0.2 C. **e** Long cycle performance at 2 C.

LLZTO-KF pellet. Then a 50 nm potassium fluoride (KF) layer was deposited on a single side or both sides of the LLZTO pellets by vacuum thermal evaporation. The deposition process is described in Fig. S2. Briefly, KF powder was put into a tungsten (W) heating boat, the deposition velocity was set as 0.5 Å/s, and the base pressure was kept below $2 \times 10^{-3}$ Pa. The deposition velocity and thickness of the KF film were estimated by SEM images and EDS mappings (Fig. S28 in Supplementary information).

### Fabrication of NCM electrodes
$LiNi_{0.8}Co_{0.1}Mn_{0.1}O_2$ (NCM) powder (Single crystal, Kelude), Super P powder (ECP-600JD, Lion Specialty Chemicals) and polyvinylidene fluoride (PVDF) powder (Kejing) were mixed well with a weight ratio of 8:1:1, and then a moderate amount of N-Methyl-2-pyrrolidinone (NMP) was added to disperse the mixture and form a homogeneous slurry by using a mixer (Thinky, ARE−310). The product was cast-coated onto a carbon-coated aluminum foil with a scraper, and the foil was transferred into a vacuum oven at 100 °C for 12 h. The drying aluminum foil was cut into small disks with a diameter of 8 mm as a positive electrode in Li|LLZTO|NCM cells.

### Materials characterization
The crystalline phase analysis of powder and pellets was performed using X-ray diffraction (XRD, Bruker D8 Advance, Cu Kα radiation), GI-

XRD (Bruker D8 Discover, Cu Kα radiation), ranging from 10° to 80°. Synchrotron radiation GIXRD experiments were performed on beamline BL02U2 with an X-ray wavelength of 0.85517 Å at the Shanghai Synchrotron Radiation Facility (SSRF). The morphology and microstructure were observed by scanning electron microscopy (SEM, JEOL, JSM-7800F). Element mappings in SEM were obtained from an energy-dispersive spectrometer (EDS, OXFORD INSTRUMENT). Surface chemical composition analysis was conducted by X-ray photoelectron spectroscopy (XPS, Thermo Fisher, ESCALAB 250Xi) and time of flight secondary ion mass spectrometry (TOF-SIMS, ION-TOF GmbH, TOF SIMS 5).

### Electrochemical measurements

Ionic conductivity and electronic conductivity were measured by the AC impedance method and DC polarization method with an electrochemical workstation (Bio-Logic, VMP-300) refer to our previous reports[40]. Au layer was sputtered on both sides of LLZTO pellets as Li-ion blocking electrodes. Electrochemical impedance spectroscopy (EIS) was performed with an AC perturbation signal of 10 mV, ranging from 7 MHz to 0.1 Hz. Direct current (DC) polarization was carried out by applying a DC voltage of 1 V and recording the current-time data until reaching a steady state (~5 h). Li|LLZTO|Li symmetric cells with/without KF layer were assembled using CR2032 coin-type cells, where the Au layer was sputtered on the LLZTO surface to improve the wettability of melted lithium metal, and the thickness of Li metal is >100 μm. Critical current density (CCD) test and long cycle test were performed in Li|LLZTO|Li symmetric cells by the LAND battery testing system and Neware battery testing system. Li|LLZTO|NCM half cells consisted of molten lithium negative electrode (Adamas-beta®), LLZTO SSEs and NCM positive electrode. For full cell assembly, various ionic liquids were adopted to wet LLZTO|Positive electrode interface. Lithium bis(trifluoromethanesulfonyl)imide (LiTFSI) was dissolved into the N-Methyl-N-propyl piperidinium bis(trifluoromethanesulfonyl)imide (PP13-TFSI), 1-butyl-1-methyl pyrrolidinium bis(trifluoromethanesulfonyl)imide (PY14-TFSI) and 1-butyl−3-methyl imidazolium bis(trifluoromethylsulfonyl)imide ionic (C₄mim-TFSI) liquid (IL), respectively. 3 μL ionic liquid was added to wet the positive electrode interface between LLZTO SSEs and NCM positive electrodes. Besides, 1 M liquid electrolyte was used to wet the interface (1 M LiPF₆ in EC/DMC (v/v = 1/1) with 2% VC, EC is ethylene carbonate, DMC is dimethyl carbonate, VC is vinylene carbonate). Rate capacity tests and long cycle tests were performed at room temperature (25 °C) and 60 °C on a battery testing system (LAND, CT2001A) with a cutoff voltage of 2.8–4.3 V. The average coulombic efficiency was calculated by cycle number and capacity retention. For example, the cells showed a capacity retention of 72.5% after 3500 cycles. Therefore, the average coulombic efficiency was $\sqrt[3500]{0.725} = 0.9999$ (99.99%).

### COMSOL simulation

The simulation results were obtained from COMSOL software with "Lithium-Ion Battery interface". The charge balance model was a "Single ion conductor". A simplified cell model was used to simulate the electric field distribution at the interface between the Li negative electrode and LLZTO solid-state electrolyte with COMSOL Multiphysics software. In the $4 \times 3 \mu m^2$ 2D geometry models (Fig. S29), the top area was LLZTO with or without electron-blocking and lithiophilic layer, and the bottom area was the Li negative electrode. The electron blocking and lithiophilic layer could improve the interface reaction uniformly. Meanwhile, the control model without a buffer layer causes a heterogeneous reaction at the interface. In the simulated deposition process, the ionic conductivity is $5 \times 10^{-4}$ S cm$^{-1}$, the applied average current density is 0.1 mA cm$^{-2}$, and the color bar denotes the current density. And more details are shown in Table S5 in the supplementary information. Electrochemical reaction kinetics at the

electrode–electrolyte interface could be described by the Butler–Volmer equation:

$$i_{loc} = i_0 \left( \exp\left(\frac{\alpha_a F \eta}{RT}\right) - \exp\left(\frac{-\alpha_c F \eta}{RT}\right) \right) \quad (4)$$

where $i_{loc}$ is the actual exchange current density, $i_0$ represents exchange current density, $\alpha_a$ is the anodic charge transfer coefficient and $\alpha_c$ is the cathodic charge transfer coefficient, $\eta$ is the overpotential, $T$ is the system temperature, $F$ is the ideal gas constant.

### DFT calculation

Density functional theory (DFT) calculations were calculated in the Vienna ab initio simulation package (VASP)[68,69]. The electron–electron interactions were treated by the projector augmented plane-wave (PAW) method. The exchange and correlation part of the density functional was calculated within the generalized gradient approximation (GGA) of Perdew–Burke–Ernzerhof (PBE)[70,71]. The structural and electronic properties were calculated in the LLZTO bulk, LLZTO slab, KF slab and LLZTO/KF interface. The plane-wave cutoff energy of 400 eV and uniform G-centered $k$-points meshes with a resolution of $2\pi \times 0.04$ Å$^{-1}$ were employed. The computational details are listed in Table S6 in the supplementary information. The interfacial formation energy ($E_{interface}$) of the LLZTO/KF interface was computed by the following equation:

$$E_{interface} = \frac{E_{LLZTO+KF} - E_{KF\ slab} - E_{LLZTO\ slab}}{S} \quad (5)$$

where $E_{interface}$ is the interfacial formation energy, $E_{LLZTO+KF}$ is the total energy of LLZTO|KF interface, $E_{KF\ slab}$ and $E_{LLZTO\ slab}$ are the total energies of KF slab and LLZTO slab, respectively, $S$ is the interfacial area. The energy barriers of Li ion migration were calculated by the nudged elastic band method.

## Data availability

The authors declare that the data supporting the findings of this study are available within the paper and its supplementary information files. Source data are provided with this paper.

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

## Acknowledgements

The authors gratefully acknowledge financial support from the National Key Research and Development Program (2019YFA0210600 to W.L. and T. Lin). The Center for High-resolution Electron Microscopy (CℏEM), Shanghai Science and Technology Plan (21DZ2260400 to Y.Y.), Double First-Class Initiative Fund of ShanghaiTech University (W.L.) and National Natural Science Foundation of China (No. 52203122 to X.C.) were also acknowledged for support. Y.C. acknowledges the support from the National Natural Science Foundation of China (51972206) and the Shanghai Municipal Science and Technology Commission (21ZR1422500).

## Author contributions

C.Z. conceived the experiment and carried out data analysis. W.L. supervised all aspects of the research. J.Y. assisted in material preparation and data analysis, Y.L. acquired the XPS spectrum and assisted in data analysis, X.C., Y.Z., Y.Y., Q.M., T. Li, and T. Lin assisted in data analysis, T.G. and Y.H. assisted in material preparation, Y.C. performed the DFT simulations, C.Z. and W.L. wrote this paper. All the authors discussed the results and commented on the manuscript.

## Competing interests

The authors declare no competing interests.
