## [Peer Review File · Nature Communications]

REVIEWER COMMENTS

Reviewer #1 (Remarks to the Author):

The article entitled Extreme electron-blocking interface for garnet-based solid-state lithium-metal 1 batteries with superior long lifespan can be considering for publication in Nature Communications after major revision.

Several aspect must be clarified:

a) Some references about interlayer must be cited in the introduction: a) Energy Storage Materials 53, 2022, 899-908 b) Advanced Functional Materials, 2023, 221019; c) ACS Energy Letters 8(10), 2023, 4016–4023

b) The role of KF is unclear: KF must react with lithium to allow a lithium ion diffusion. The paper cannot be published until a clear mechanism is proposed: I suggest to put in contact a KF powder with Li-metal in glove box: KF should cover Li's surface. The authors should check the final formed phase by Grazing Angle XRD and a cross section SEM/EDX mapping.

c) The formation of F-rich particles in Figure 4 seems bigger than 40 nm: why?

d) The formation of F rich particles suggests the existence of a reaction between Li and KF. The authors should prepare LLZO with different % of KF as co-sintering agent (5 and 10%) and compare the cycling results with sputtered KF-LLZO sample (symmetrical Li-Li cells and half cell with NMC). KF will be probably segregated at the grain boundaries, it's possible to observe an improvement similar to sputtering method

e) The authors should consider the influence of Li₂CO₃ on LLZO performance: a) Advanced Materials 35, 2023, 2208951; b) ACS Applied Energy Materials 3 (4), 2020, 3415-3424; c) Advanced Functional Materials 31, 2021, 2103716

f) In Figure 4 the authors should add cross section SEM image and EDX mapping of Li-KF-LLZO interface after cycling

g) In Figure 6 the authors should add Li-LLZO-NMC cell as reference (without KF interlayer)

h) The author should increase the C-rate up to 1C and 2C and report the final data in the manuscript.

Reviewer #2 (Remarks to the Author):

A thin KF layer is deposited on the surface of an LLZTO pellet, and it is demonstrated that the interfacial behavior between Li metal and LZTO can be meaningfully improved. I do not recommend the publication of this manuscript in its current form for the following reasons: (1) some essential data are missing to support the core concept of this work, and (2) some of the provided electrochemical data do not seem to be appealing.

1. It is mentioned that a 50 nm thick KF layer is deposited on the LLZTO pellet, but no analytical data are provided to support it. Clear evidence should be provided to show that the thickness is approximately 50 nm, and that the layer is dense and uniform.

2. The XPS data in Fig. 2 indirectly indicate that the KF layer is flawed. Li on the substrate should not be detected if a well-formed, 50 nm thick KF layer is in place. The presence of a Li peak in XPS tells us that the coverage is not complete.

3. Cross-sectional SEM images are provided only after cycling. Specific cycling information, such as current density and the number of cycles, is also missing. To demonstrate the behavior of the KF layer, images taken before cycling should be provided and compared with those obtained after cycling.

4. The role of the KF layer is suggested to prevent the formation of Li dendrites, but what we observe is that void formation, rather than dendrites, is mitigated. More explanation is needed.

5. The COMSOL simulation does not seem relevant to the core concept of this work. Physical and transport properties of KF are not reflected in the simulation. It only simulates the influence of voids at the interface, which is highly predictable. I do not believe this simulation study provides valuable insight or demonstrates anything meaningful.

6. The Li ion conductivity of the KF layer should be provided or measured. Additionally, the added resistance caused by the 50 nm thick KF layer should be calculated, and its impact on overall cell performance should be discussed.

7. LiTFSI in C4mim is chosen as the catholyte, but its performance is not adequate for use in commercial cells. The cathode loading is approximately 2.5 mg/cm^2 , corresponding to an areal capacity of 0.5 mAh/cm^2 or lower. Therefore, 0.5 C would be 0.25 mA/cm^2 or lower. The fact that only 80% of the capacity is obtained at such low current density and low loading suggests that this catholyte is not suitable. In typical Li-ion batteries, the areal capacity is higher than 3 mAh/cm^2 , and more than 90% of capacity is recovered at 0.5 C (compared to 0.1 C capacity).

8. There are numerous grammatical errors and incorrect word usage in the manuscript. Here are a few examples, but there are many more.

"Garnet oxide"

"physicochemical (.....)"

"security risks"

"In order to comparing"

"for fully reaction"

"after nature cooling"

"the Extreme electron-blocking"

"were compared of"

Reviewer #3 (Remarks to the Author):

Authors demonstrate that a highly effective interlayer composed of potassium fluoride (KF) can inhibit lithium dendrite growth in garnet oxide (LLZTO), resulting in an ultralong cycle life of over 3000 hours at 0.2 mA cm^{-2} and over 350 hours at 0.5 mA cm^{-2} in room temperature. The paper also discusses the mechanism behind the KF interlayer's ability to block electro migration and inhibit dendrite growth, as well as the potential applications of this approach in the development of solid-state lithium-metal batteries. This paper presents a novel approach to developing an extreme electron-blocking interface for garnet-based ASSBs. The manuscript is well written systematically; however, I found some few doubts. After addressing, it will be published in high-quality journal.

Q1) Were the electrochemical impedances measured not only in full-cell configurations but also in electron-blocking and ion-blocking cell configurations at various temperatures, ranging from room temperature to 65°C?

Q2) The crystallinity of the KF phase is ambiguous. The XRD peaks matched with PDF#36-1458, but Figure 3a showed PDF#85-1314 after Li-KF reactions. Does the deposited KF on LLZTO exhibit crystallinity and remain stable after multiple cycles?

Q3) The COMSOL simulation displays the electrical current distribution between electrolytes and electrodes. However, technically, electrical current density cannot flow through the electrolyte phase. Can the geometrical differences due to voids support the effect of KF on LLZTO? The focusing of current density near the narrow interface seems to result from geometrical effects, not the KF interface. More comprehensive details are required for clarity.

Q4) In reference 47, Huo et al. in Nat Commun (2021) proposed a similar concept of an electron-blocking layer between LLZTO/Li. To validate the role of electron-blocking by PAA, they demonstrated depth profiles of TOF-SIMS for LLZTO/PAA pellets and performed electronic calculations using DFT simulations. In this study, although the authors suggested the structure of KF as PDF#85-1314, the details of the structure and the origin of electron-blocking by KF itself are not clear.

Q5) Potassium fluoride is thermodynamically less stable than lithium fluoride. If excess lithium metal grows, KF may be replaced by LiF. The thermodynamic enthalpy for the reaction of KF with Li to form LiF and K is -0.507 eV or -49.055 kJ mol⁻¹.

Q6) The KF thin layer shows reduced electronic conductivity, increasing resistance and overpotential in Li symmetric cells. Why did KF-LLZTO show lower overpotential in the early cycle stages compared to LLZTO, despite the higher overpotentials observed under 0.2 mA cm⁻² (0.1 mAh cm⁻²) in both LLZTO and KF-LLZTO cells?

Title: Extreme electron-blocking interface for garnet-based solid-state lithium-metal batteries with superior long lifespan

Corresponding author: Wei Liu

Manuscript ID: NCOMMS-23-48483-T

Response to the reviewers' comments

First of all we would like to thank the reviewers for their time and effort in reviewing this manuscript. The comments are all valuable and constructive for revising and improving our paper, as well as the important guiding significance to our researches. We have provided the point-by-point responses below **in blue text** while keeping the reviewer's comments in black.

REVIEWER COMMENTS

Reviewer #1 (Remarks to the Author):

The article entitled Extreme electron-blocking interface for garnet-based solid-state lithium-metal batteries with superior long lifespan can be considering for publication in Nature Communications after major revision. Several aspects must be clarified:

Response: We thank the reviewer's comments and constructive suggestion. We have designed corresponding experiment, modified some descriptions and details, and added more discussions in the revised manuscript. The detailed answer is given as bellow.

Question 1: Some references about interlayer must be cited in the introduction: a) Energy Storage Materials 53, 2022, 899-908 b) Advanced Functional Materials, 2023, 2210192; c) ACS Energy Letters 8(10), 2023, 4016–4023

Response: Thanks for this comment. We have added relevant discussions and reference in the revised Main Text as follow: (Page 2)

“Moreover, the poor interface contact between Li and LLZO caused the high interfacial resistance and heterogeneous current distribution, which were to blame for lithium dendrite formation. Some approaches were report to alleviate the issues, such as lithium alloys anode, stable Li-ion conducting layer, and coating metal alloys by melt-quenching process²⁷⁻²⁹. These interlayers could reduce the interface

resistance and improve the cycle stability.”

Question 2: The role of KF is unclear: KF must react with lithium to allow a lithium ion diffusion. The paper cannot be published until a clear mechanism is proposed: I suggest to put in contact a KF powder with Li-metal in glove box: KF should cover Li's surface. The authors should check the final formed phase by Grazing Angle XRD and a cross section SEM/EDX mapping.

Response: Thanks for the constructive suggestions. We have added corresponding experiment and characterization analysis. We have confirmed that the reaction mechanism is **Reaction 1**, the detailed experiments and supporting evidences are as follows:

For our solid-state lithium metal cells, the lithium anode was prepared by dipping the LLZTO-KF pellet into molten lithium, where lithium metal was tightly bonded to pellet and it was difficult to separate them apart to observe the phase change of the interlayer material. In order to observe the reaction product of Li and KF, we have designed two kinds of experiments as follow: (1) KF powder react with melted Li metal, and (2) KF green pellet react with melted Li metal.

(1), put some KF powder into melted Li metal on stainless steel substrate with 300 °C hotplate for fully reaction.

(2), make a KF pellet in a stainless-steel die (d=12.7 mm) under a uniaxial pressure of 300 MPa, and put the KF green pellet into melted Li metal for 30 min. Then take out the KF pellet from melted Li (KF surface was covered by Li metal), and put on the 300 °C hotplate for fully reaction.

Then, XRD and grazing incidence XRD (GI-XRD) were carried out with the two samples. For sample 1# (KF powder into melted Li metal), strong XRD peaks of Li (PDF#15-0401) and KF (PDF#85-1314) were collected as shown in Figure R1a (black line). More importantly, the weak XRD peaks of reaction product LiF (PDF#45-1460) were observed successfully. For a clearer detection of the reaction products, as suggested by this reviewer, GI-XRD was performed with an incidence angle of 1°. And stronger peaks of LiF were observed clearly (Figure R1b, black line). For sample 2# (melted Li on KF pellet), strong and clear peaks of *Li, KF and LiF* were detected from XRD and GI-XRD patterns (Figure R1, blue lines).

Figure R1. The XRD (a) and GI-XRD (b) patterns with an incidence angle of 1° of Li-KF reaction product under Kapton film.

In addition, the phase structure of LLZTO-KF|Li sample (dipping LLZTO-KF pellet into melted lithium fleetly) was verified by synchrotron radiation GI-XRD with different incidence angle of 0.2° , 0.5° and 1° (Figure R2a). Thanks to the high energy and high resolution of the synchrotron radiation X-ray, the phase structure of interlayer with very low content has been successfully detected. Figure R2b displayed the enlarged synchrotron radiation GI-XRD patterns, except the LLZTO substrate, *the peaks of KF (28.9° , 33.5° and 48.1°), LiF (38.7° and 45°) and Li (36.2°) were observed clearly.*

Figure R2. Synchrotron radiation GI-XRD patterns (a) and the enlarged picture (b) of LLZTO-KF|Li sample at the incidence angle of 0.2° , 0.5° and 1° .

Moreover, to verify the reaction between the KF layer and Li clearly and easily, depth-profiling X-ray photoelectron spectroscopy (XPS) analysis was performed to obtain the composition information in different depth by ion etching. We firstly thermally evaporate ~100 nm lithium metal on the substrate, and then thermally evaporate a ~50 nm KF layer on the Li metal surface in glovebox, as shown in Figure R3a inset. Before etching, only K-F signal was clearly collected on the top layer (KF layer) surface of F 1s regions. And classical K-F bond was observed in K 2p regions, while no markedly Li signal was detected on the sample surface. With the etching depth increasing, the signal of generated LiF began to appear and gradually increased according to the XPS spectrum of F 1s and Li 1s. Moreover, the K-O signal was collected at 50 nm depth and it became stronger at 100 nm (Figure R3b), where the K-O signal come from the generated K metal (K is active metal, always showing K-O signal in XPS test). For Li 1s regions, as displayed in Figure R3c, lithium metal signal began to appear when etching depth arriving Li metal layer (50 nm) and lithium metal peak occupied a higher proportion when arriving the interior lithium layer. Combining the characterization results of XRD, GI-XRD and depth-profiling XPS, we can prove that the reaction mechanism is **Reaction 1** discussed above.

Figure R3. XPS spectrums of Substrate|Li|KF samples. (a-c) XPS spectrum of F 1s, K 2p and Li 1s regions with various etching depth of 0, 20, 50 and 100 nm, respectively.

Specially, we found more intuitive evidence that the marked reaction process could be observed directly with the naked eye. We thermally evaporate ~100 nm lithium metal on LZTO-KF pellet, the surface was

covered by gray lithium metal. Then put a piece of LLZTO-KF|Li pellet on hotplate for 200 °C heat treatment for 10 min. Finally, we were surprised to find that the gray lithium metal on the surface nearly disappeared after heat treatment. Figure R4a exhibited the photo of LLZTO-KF|Li pellet with/without heat treatment. XPS test was performed to analyse the surface chemical constituents. As shown in Figure R4b-d, before heat treatment, the Li₂O and Li₂CO₃ signals were obtained from the thermally evaporated lithium metal surface. Weak adsorbent F 1s peak was collected, but no any KF was found according to XPS spectrum of F 1s and K 2p regions. As the contrast, the gray lithium nearly disappeared after heat treatment. In addition to the unreacted Li metal and KF, the generated LiF and K were also observed on the sample surface. This finding proves that heat treatment could promote **Reaction 1**.

Figure R4. (a) The photos of LLZTO-KF|Li pellet with/without heat treatment. (b-d) XPS spectrum of Li 1s, F 1s and K 2p regions of LLZTO-KF|Li pellet with/without heat treatment, respectively.

Furthermore, we refer to the two-phase diagram of LiF-Li and KF-K reported by A. S. Dworkin *et al.* ^[1], LiF does not dissolve into melted Li metal below 845 °C and KF does not dissolve into melted K metal

below 850 °C (Figure R5). For our LLZTO-KF|Li interface, the forming LiF can prevent the further reaction between KF and Li at the interface and the forming K does not affect the insulation of the interlayer. Therefore, the ionic-conducting and electron-blocking layer consisted of LiF and KF was *in situ* generated after LLZTO-KF reacting with melted lithium metal.

Figure R5. Liquid metal-salt phase equilibria in the alkali metal-fluoride systems (LiF-Li and KF-K systems) ^[1].

According to this reviewer's suggestion, the cross-section SEM image and EDS mappings of Li-KF reaction product (KF powder into melted Li metal) have also been added (Figure 6R). In EDS mappings, the signals of F and K enriched in the Li metal surface region, which may correspond to unreacted KF and forming LiF according to the results above (Figure R1-R5).

Figure R6. The cross-section SEM image (a) and EDS mappings of O (b), F (c) and K (d) elements of Li-KF reaction product.

Finally, the results, discussions and figures were modified in the revised Main Text in blue text according to this comment. (Page 5-6)

“Next, the reaction between KF layer and lithium metal was studied scientifically. For our solid-state lithium metal cells, the lithium anode was prepared by dipping the LLZTO-KF pellet into molten lithium, where lithium metal was tightly bonded to pellet and it was difficult to separate them apart to observe the phase evolution of the interlayer material. In order to observe the reaction product of Li and KF, we have designed two kinds of experiments. (1) KF powder react with melted Li metal: put some KF powder into melted Li metal on stainless steel substrate with 300 °C hotplate for fully reaction. (2) KF green pellet react with melted Li metal: make a KF pellet in a stainless-steel die under a uniaxial pressure of 300 MPa, and put the KF green pellet into melted Li metal for 30 min. Then take out the KF pellet from melted Li (KF surface was covered by Li metal), and put on the 300 °C hotplate for fully reaction. Then, XRD and grazing incidence XRD (GI-XRD) were carried out with the two samples. For sample 1# (KF powder into melted Li metal), strong XRD peaks of Li (PDF#15-0401) and KF (PDF#85-1314) were collected as shown in Figure S7a (black line). More importantly, the weak XRD peaks of reaction product LiF (PDF#45-1460) were observed successfully. For a clearer detection of the reaction products, GI-XRD was performed with an incidence angle of 1°. And stronger peaks of LiF were observed clearly (Figure S7b, black line). For sample 2# (melted Li on KF pellet), strong and clear peaks of Li, KF and LiF were

detected from XRD and GI-XRD patterns (Figure S7a blue line, and Figure 2a). In addition, the phase structure of LLZTO-KF/Li sample (dipping LLZTO-KF pellet into melted lithium fleetly) was verified by synchrotron radiation GI-XRD with different incidence angle of 0.2°, 0.5° and 1° (Figure S8 and Figure 2b). Thanks to the high energy and high resolution of the synchrotron radiation X-ray, the phase structure of interlayer with very low content has been successfully detected. Figure 2b displayed the enlarged synchrotron radiation GI-XRD pattern at 1°, except the LLZTO substrate, the peaks of KF (28.9°, 33.5° and 48.1°), LiF (38.7° and 45°) and Li (36.2°) were observed clearly.”

“Moreover, to verify the reaction between the KF layer and Li clearly and easily, depth-profiling X-ray photoelectron spectroscopy (XPS) analysis was performed to obtain the composition information in different depth by ion etching. We firstly thermally evaporate ~100 nm lithium metal on the substrate, and then thermally evaporate a ~50 nm KF layer on the Li metal surface in glovebox, as shown in Figure 2c inset. Before etching, as shown in Figure 2c-e, classical K-F signal was clearly collected on the top layer (KF layer) surface of F 1s and K 2p regions, while no markedly Li signal was detected on the sample surface. With the etching depth increasing, the signal of generated LiF began to appear and gradually increased according to the XPS spectrum of F 1s and Li 1s. Moreover, the K-O signal was collected at 50 nm depth and it became stronger at 100 nm (Figure 2d), where the K-O signal come from the generated K metal (Notably, K is the active metal, always showing K-O signal in XPS result). For Li 1s regions, as displayed in Figure 2e, lithium metal signal began to appear when etching depth arriving Li metal layer (50 nm) and lithium metal peak occupied a higher proportion when arriving the interior lithium layer (100 nm). More intuitive evidence and discussion were given in Figure S9 and Note S1.

KF can react with molten Li metal at high temperatures according to Reaction 1:

According to the LiF-Li and KF-K phase diagrams as shown in Figure S10⁴¹, the forming LiF does not dissolve into Li and forming K does not dissolve into KF. For our LLZTO-KF/Li interface, during assembling LLZTO-KF with Li metal anode, KF reacted with molten Li metal partially, and the forming LiF could prevent the further reaction between KF and Li at the interface. And, the forming K with very low melting pointing might diffuse into molten Li metal during reaction, which did not affect the insulation of the interlayer. Therefore, the results of XRD, GI-XRD and depth-profiling XPS supported the above reaction mechanism. The in-situ reaction of KF and Li metal improves the lithiophilic ability and reduces the interface impedance, resulting in excellent electrochemical performance.”

Figure 2 in revised Main Text. Reaction mechanism of Li and KF buffer layer. (a) The GI-XRD patterns with an incidence angle of 1° of Li-KF reaction product under Kapton film. (b) Synchrotron radiation GI-XRD patterns at the incidence angle of 1° . (c-e) XPS spectrum of F 1s (c), K 2p (d) and Li 1s (e) regions with various etching depth of 0, 20, 50 and 100 nm, respectively. The inset in (c) is the structure schematic diagram of Substrate|Li|KF samples.

Figure S7 in revised supplementary information. The XRD (a) and GI-XRD (b) patterns with an incidence angle of 1° of Li-KF reaction product under Kapton film.

Figure S8 in revised supplementary information. Synchrotron radiation GI-XRD patterns (a) and the enlarged picture (b) of LLZTO-KF|Li sample at the incidence angle of 0.2° , 0.5° and 1° .

Figure S9 in revised supplementary information. (a) The photos of LLZTO-KF|Li pellet with/without heat treatment. (b-f) XPS spectrum of Li 1s, F 1s, K 2p, La 3d and Zr 3d regions of LLZTO-KF|Li pellet with/without heat treatment, respectively.

Figure S10 in revised supplementary information. Liquid metal-salt phase equilibria in the alkali metal-fluoride systems (LiF-Li and KF-K systems).^[1]

Question 3: The formation of F-rich particles in Figure 4 seems bigger than 40 nm: why?

Response: We thank the reviewer's comment. We mentioned in Main Text: "In our work, KF with 50 nm thickness was deposited on Li₂CO₃ contaminant free LLZTO pellet surface". It needs to be clarified that the SEM images in **previous Figure 4** is the data for LLZTO-KF|Li interface and LLZTO|Li interface **after cycling**. And the images before cycle have been added in **Question 6** and revised manuscript. After cycles, the thickness of F-rich interface layer is increased to ~250 nm, which may be due to the following two reasons:

- (1) the formation of F-rich particles due to the *in-situ* reaction between Li and KF.
- (2) the cross-section of sample is not flat enough with ups and downs, so the EDS signals of the interlayer show a discontinuous green line in **Figure 3e (revised Main Text)**.

Question 4: The formation of F rich particles suggests the existence of a reaction between Li and KF. The authors should prepare LLZO with different % of KF as co-sintering agent (5 and 10%) and compare the cycling results with sputtered KF-LLZO sample (symmetrical Li-Li cells and half-cell with NMC). KF will be probably segregated at the grain boundaries, it's possible to observe an improvement similar to sputtering method.

Response: Thanks for this suggestion. We have prepared the LLZTO pellets with 5% and 10% KF powder as sintering agent, named as LLZTO-5% KF and LLZTO-10% KF. Mix LLZTO powder with different percent KF powder and grind well by using a mortar and pestle. Then make the pellet in a stainless-steel die (d=12.7 mm) under a uniaxial pressure of 300 MPa, and sinter the samples with the same process as LLZTO samples (1250°C, 40 min). Figure R7 displayed the electrochemical performance of LLZTO-5% KF and LLZTO-10% KF. After sintering process, LLZTO-5% KF pellets became denser with a diameter of ~11.1 mm, and showing an ionic conductivity of $1.1 \times 10^{-4} \text{ S cm}^{-1}$ (lower than that of LLZTO and LLZTO-KF). While there was almost no change of LLZTO-10% KF pellets before and after sintering process, with an extreme low ionic conductivity of $0.07 \times 10^{-4} \text{ S cm}^{-1}$. Then critical current density (CCD) test was employed to evaluate the LLZTO-5% KF pellets, and the samples displayed a CCD of 0.3 mA cm⁻² (Figure R7c) Notice that no CCD data was acquired from LLZTO-10% KF samples because of the incompact pellets and low ionic conductivity. Figure R7d was the electrochemical performance comparison chart of LLZTO-KF (discussed in revised main text), LLZTO-5% KF and LLZTO-10% KF. According to above results, we believe KF is not a good sintering aid candidate for compacting LLZO pellet or the percentage of KF needs further optimizing. In addition, a uniform covering of KF on LLZO

surface is required for improving CCD.

Figure R7. Electrochemical performance of LLZTO-5% KF and LLZTO-10% KF. (a, b) Nyquist plots and the enlarged plots of LLZTO-5% KF and LLZTO-10% KF. The insets are the photos of samples after sintering process. (c) CCD profiles at time-constant mode of LLZTO-5% KF. (d) Electrochemical performance comparison chart of LLZTO-KF, LLZTO-5% KF and LLZTO-10% KF.

Question 5: The authors should consider the influence of Li₂CO₃ on LLZO performance: a) Advanced Materials 35, 2023, 2208951; b) ACS Applied Energy Materials 3 (4),2020, 3415-3424; c) Advanced Functional Materials 31, 2021, 2103716.

Response: Thanks for this comment. We have added the relative discussion about the effect of Li₂CO₃ and cited the mentioned literatures in the revised Main Text. In our previous work [2], the effects of Li₂CO₃ were studied scientifically. It is approved that: 1) Li₂CO₃ is lithiophobic and the Li₂CO₃-free LLZTO is intrinsically lithiophilic; 2) Li₂CO₃ can form and accumulate on the surface of LLZTO when storing in air, which will reduce the Li wettability and lead to large interfacial impedances; 3) Lithiophilic

interlayers can improve the Li wettability of LLZTO pellets even when Li_2CO_3 layer is already formed, but cannot reduce the areal specific resistance. In this work, the surface of LLZTO pellets was polished carefully, and then KF layer was thermally evaporated onto the LLZTO surface in Ar glovebox. Therefore, there is nearly no Li_2CO_3 contaminant on the surface of our LLZTO-KF pellet.

Finally, the discussions were modified in the revised Main Text in blue text according to this comment. (Page 2 and 12)

“In addition, LLZO are easily susceptible to Li_2CO_3 formation from exposure to water vapor and carbon dioxide (atmospheric environment). The forming Li_2CO_3 will cause the lithiophobic interface with poor interface contact and enlarged interface resistance²⁷⁻³⁰. Moreover, the poor interface contact between Li and LLZO caused the high interfacial resistance and heterogeneous current distribution, which were to blame for lithium dendrite formation. Some approaches were reported to alleviate the issues, such as lithium alloys anode, stable Li-ion conducting layer, and coating metal alloys by melt-quenching process³¹⁻³³. These interlayers could reduce the interface resistance and improve the cycle stability.”

“The synthetic LLZTO powder and 2 wt% nano $\gamma\text{-Al}_2\text{O}_3$ (Macklin) were fully ground by a mortar and pestle, where Al_2O_3 was used as sintering additive for dense LLZTO pellet. Then ~0.5 g fine powders were transferred into a stainless-steel die ($d=12.7$ mm) and pressed under a uniaxial pressure of 300 MPa for 2 min. These green pellets were sintered in MgO crucible at 1250 °C for 40 min in air with a heating rate of 5 °C min^{-1} . The LLZTO pellets were grinded and polished with sandpaper and abrasive paste, and ethanol was used to clean these pellets by ultrasonic wave. And then the fresh pellets were transferred to glovebox with Ar immediately. Therefore, there is nearly no Li_2CO_3 contaminant on the surface of our LLZTO-KF pellet. Then 50 nm potassium fluoride (KF) layer was deposited on single side or both sides of the LLZTO pellets by vacuum thermal evaporation. The deposition process was described by Figure S2 (Supplementary information). Briefly, KF powder was put into tungsten (W) heating boat, the deposition velocity was set as 0.5 Å/s, and the base pressure was kept below 2×10^{-3} Pa. The deposition velocity and thickness of KF film were estimated by SEM images and EDS mappings (Figure S27 in supplementary information).”

Question 6: In Figure 4 the authors should add cross section SEM image and EDX mapping of Li-KF-LLZO interface after cycling.

Response: Thanks for this comment. There may be some misunderstanding about the SEM images in Figure

4 in previous Main Text. It needs to be clarified that the SEM images in Figure 4 is the data after cycling. And the images before cycling have been added as shown in Figure S11. Both LLZTO|Li interface and LLZTO-KF|Li interface are tight before Li plating and stripping process.

Finally, the results, discussions and figures were modified in the revised Main Text in blue text according to this comment. (Page 7)

“Before Li plating and stripping process, both of LLZTO|Li interface and LLZTO-KF|Li interfaces are tight as shown in Figure S11.”

Figure S11 in revised supplementary information. Cross-section SEM images of LLZTO|Li and LLZTO-KF|Li interfaces before cycling.

Question 7: In Figure 6 the authors should add Li-LLZO-NMC cell as reference (without KF interlayer).

Response: Thanks for point it out. We have added the Li|LLZTO|NMC cells as reference (Figure 6 in revised Main Text). And the results, discussions and figures were modified in the revised Main Text in blue text according to this comment, as follows: (Page 10)

“Moreover, electrochemical performance of garnet-based solid-state Li|NMC cells with 1 M C₄mim was measured. Charge-discharge curves of Li|KF-LLZTO-IL|NMC cells at various C-rate were displayed in Figure S21. With the increasing of C-rate, the charging-discharging curves maintain the good shapes ranged from 2.8-4.3 V with low polarizations at 0.05C, 0.1C, 0.2C, 0.5C and 1C (1C=190 mAh g⁻¹). The rate test result was exhibited in Figure 6b (mass loading of NCM is ~2.5 mg cm⁻²). At the C-rate of 0.05C, an activation process occurred at first 5 cycles, presenting the increasing specific capacity from 183.4 to

192.4 mAh g⁻¹. And the specific capacity of 186.8, 177.6, 147.8, 117.5 and 63.3 mAh g⁻¹ were realized at 0.1C, 0.2C, 0.5C, 1C and 2C, respectively. When returned to 0.2C, a high specific capacity of 176.2 mAh g⁻¹ was observed, 96.1% of that at 0.05C. For the cells with bare LLZTO, high specific capacity of 193.2 and 179.1 mAh g⁻¹ were realized at 0.05C and 0.1C. However, the capacity decayed rapidly after 0.5C and was short circuit at 2C. As for long cycle test, for Li|NCM cells with LLZTO-KF, the high specific capacity of 178.2 mAh g⁻¹ was realized, with a high capacity retention of 82.0% at 0.2C after 300 cycles as shown in Figure 6c. As a contrast, Li|LLZTO-IL|NCM cells show an initial specific capacity of 139.4 mAh g⁻¹ at 0.2C and decay to 117 mAh g⁻¹ after 221 cycles (Figure S22). Also, high mass loading NCM of ~4 mg cm⁻² was adopted and cycled at 0.2C. As displayed in Figure S23, the cells with LLZTO-KF have a high initial specific capacity of 145.9 mAh g⁻¹, with a capacity retention of 83.8% after 125 cycles. In consideration of the CCD of LLZTO SSEs and cell polarizations at high current density, the long cycle tests at 1C and 2C were performed with NCM cathode (mass loading is ~1 mg cm⁻²). Cells with LLZTO-KF displayed stable cycle life over 1000 cycles at 1C, showing initial specific capacity of 123.1 mAh g⁻¹ and capacity retention of 83.1% (Figure S24). Besides, the excellent capacity retention of 72.5% was realized after 3500 cycles at 2C and RT (Figure 6d), representing an average coulombic efficiency of 99.99%. As a contrast, cells with bare LLZTO showed a specific capacity of 22.9 mAh g⁻¹, and decay to lower than 10 mAh g⁻¹ after 45 cycles. For further evaluate cycle stability and electrochemical performance, the cells were tested at 2C in 60 °C thermostat. As shown in Figure S25, a high initial specific capacity of 186.1 mAh g⁻¹ was presented in Li|KF-LLZTO-IL|NCM cells, delivering the high capacity of 139.9 mAh g⁻¹ after 1000 cycles. While, Li|LLZTO-IL|NCM cells without modifying layer displayed a low initial capacity of only 121.8 mAh g⁻¹.”

Figure 6 in revised Main Text. Electrochemical performances of Li|LLZTO-IL|NCM and Li|KF-LLZTO-IL|NCM cells at RT, the ionic liquid is 1 M LiTFSI in C_{mim}-TFSI. (a) Nyquist plots of cells of Li|KF-LLZTO-IL|NCM, the inset is the equivalent circuit. (b) Rate capability. (c) Long cycle performance at 0.2C. (d) Long cycle performance at 2C and RT. (e) Cycle performance comparison chart of our work with other literature reports.

Figure S21. Charge-discharge curves of Li|KF-LLZTO-IL|NCM at various C-rate.

Figure S22. Long cycle performance of Li|LLZTO-IL|NCM cells with mass loading of $\sim 2.5 \text{ mg cm}^{-2}$ at 0.2C.

Figure S23. Long cycle performance of Li|KF-LLZTO-IL|NCM cells with mass loading of $\sim 4 \text{ mg cm}^{-2}$ at 0.2C.

Figure S24. Long cycle performance of Li|KF-LLZTO-IL|NCM cells at 1C.

Figure S25. Long cycle performance of Li|KF-LLZTO-IL|NCM and Li|LLZTO-IL|NCM cells at 2C and 60 °C.

and 60 °C.

Question 8: The author should increase the C-rate up to 1C and 2C and report the final data in the manuscript.

Response: Thanks for point it out. We have added the rate performance test of 1C and 2C in Figure 6b. And the results, discussions and figures were modified in the revised Main Text in blue text according to this comment. (Page 10)

“The rate test result was exhibited in Figure 6b (mass loading of NCM is $\sim 2.5 \text{ mg cm}^{-2}$). At the C-rate of 0.05C, an activation process occurred at first 5 cycles, presenting the increasing specific capacity from 183.4 to 192.4 mAh g^{-1} . And the specific capacity of 186.8, 177.6, 147.8, 117.5 and 63.3 mAh g^{-1} were realized at 0.1C, 0.2C, 0.5C, 1C and 2C, respectively. When returned to 0.2C, a high specific capacity of 176.2 mAh g^{-1} was observed, 96.1% of that at 0.05C. For the cells with bare LLZTO, high specific capacity of 193.2 and 179.1 mAh g^{-1} were realized at 0.05C and 0.1C. However, the capacity decayed rapidly after 0.5C and was short circuit at 2C.”

Figure 6b in revised Main Text. Rate capability of Li|KF-LLZTO-IL|NCM and Li|LLZTO-IL|NCM cells, the ionic liquid is 1 M LiTFSI in C_{mim}-TFSI.

Reference

- [1] Dworkin, A. S., H. R. Bronstein, and M. A. Bredig. "Miscibility of metals with salts. vi. lithium-lithium halide systems1." *The Journal of Physical Chemistry* 66.3 (1962): 572-573.
- [2] Chen, Shaojie, et al. "The influence of surface chemistry on critical current density for garnet

electrolyte." *Advanced Functional Materials* 32.23 (2022): 2113318.

Reviewer #2 (Remarks to the Author):

A thin KF layer is deposited on the surface of an LLZTO pellet, and it is demonstrated that the interfacial behavior between Li metal and LZTO can be meaningfully improved. I do not recommend the publication of this manuscript in its current form for the following reasons: (1) some essential data are missing to support the core concept of this work, and (2) some of the provided electrochemical data do not seem to be appealing.

Response: We thank the reviewer's comments. Also, we appreciate the reviewer's critical suggestions on the experiment methods, results, and discussions. We have conducted more detailed work and modified some descriptions and discussions in the revised manuscript in order to emphasize the scientific significance and improve the manuscript. The detailed answer is given as bellow.

Question 1: It is mentioned that a 50 nm thick KF layer is deposited on the LLZTO pellet, but no analytical data are provided to support it. Clear evidence should be provided to show that the thickness is approximately 50 nm, and that the layer is dense and uniform.

Response: Thanks for point it put. According to this comment, the SEM images and EDS mappings of LLZTO and LLZTO-KF with related discussions were added in the revised manuscript.

Figure R1 showed the cross-sectional view SEM images of LLZTO and LLZTO-KF surface with corresponding EDS mappings. The LLZTO pellet showed a smooth surface, and a thin KF layer on dense LLZTO pellet was observed clearly after thermal evaporation. To calibrate the thickness more accurately, we thermally evaporated the KF layer onto silicon wafer at the same time and conditions as on LLZTO-KF pellet, and then measured the thickness of the KF layer by cross-section SEM and EDS mapping. As shown in Figure R2, a dense and uniform KF layer was observed clearly, and the thickness is about 50 nm according to the line scan EDS spectrum and mappings.

Figure R1. The cross-sectional view SEM images of LLZTO (a) and LLZTO-KF (b) pellet with corresponding EDS mappings (c-d) of LLZTO-KF.

Figure R2. (a) Cross-section SEM image of Si-substrate|KF interface. (b) EDS spectrum of full elements. (c) Line scan EDS spectrum of Si, K and F elements from point A to point B. (d-f) EDS mappings of Si (d), F (e) and K (f) corresponding to the SEM image (a).

In addition, Figure R3a displayed SEM image of the KF thin film on Si-substrate. Notably, the area attached some impurity particles was selected for further EDS mapping test to distinguish the different

elements clearly. Figure R3b exhibited the EDS spectrum of full elements, where K element and F element were detected obviously, and the weight percentage were 2.6 wt% and 2.2 wt%, respectively. Figure R3c is the EDS mappings of C element, the high light areas indicated the impurity particles. Figure R3d-f showed the uniform element distribution of Si, F and K, signifying the uniform KF thin film on Si substrate.

Figure R3. (a) SEM image of KF thin film on Si-substrate. (b) EDS spectrum of full elements. (c-f) EDS mappings of C (c), Si (d), F (e) and K (f) corresponding to the SEM image (a).

Moreover, atomic force microscope (AFM) was employed to measure the surface morphology of LLZTO-KF pellet. Figure R4a is the AFM topography image of LLZTO-KF, and the corresponding 3D topographic image is displayed in Figure R4b, indicating the dense and uniform layer. Section analysis was given in Figure R4c, showing the surface roughness of ~25 nm. However, the LLZTO pellet (substrate) was prepared from LLZTO powders sintering, therefore the formation of grain boundary was inevitable as shown in Figure R1a-b. Meanwhile, the holes on KF layer were observed, which maybe correspond to the grain boundary and voids of LLZTO pellet.

Figure R4. (a) AFM topography image of LLZTO-KF ($10 \times 10 \mu\text{m}$). (b) 3D AFM topographic image of (a). (c) Height profiles along the yellow line in the 2D image of (a).

Finally, the results, discussions and figures were modified in the revised manuscript in blue text according to this comment. (Page 4, 12)

“Besides, cross-sectional SEM image and EDS mappings of LLZTO-KF were shown in Figure 1c, where the smooth surface and KF modifying layer were observed clearly. Moreover, atomic force microscope (AFM) was employed to measure the surface morphology of LLZTO-KF pellet. Figure S5a is the AFM topography image of LLZTO-KF, and the corresponding 3D topographic image is displayed in Figure S5b, indicating the dense and uniform layer. Section analysis was given in Figure S5c, showing the surface roughness of $\sim 25 \text{ nm}$.”

“The deposition velocity and thickness of KF film were estimated by SEM images and EDS mappings (Figure S27 in Supplementary information).”

Figure 1c in revised Main Text. Cross-sectional SEM image of LLZTO-KF and corresponding EDS mappings.

Figure S5 in revised supplementary information. (a) AFM topography image of LLZTO-KF ($10 \times 10 \mu\text{m}$). (b) 3D AFM topographic image of (a). (c) Height profiles along the yellow line in the 2D image of (a).

Figure S26 in revised supplementary information. (a) Cross-section SEM image of Si-substrate|KF interface. (b) EDS spectrum of full elements. (c) Line scan EDS spectrum of Si, K and F elements from point A to point B. (d-f) EDS mappings of Si (d), F (e) and K (f) corresponding to the SEM image (a).

Question 2: The XPS data in Fig. 2 indirectly indicate that the KF layer is flawed. Li on the substrate should not be detected if a well-formed, 50 nm thick KF layer is in place. The presence of a Li peak in XPS tells us that the coverage is not complete.

Response: Thanks for this comment. We agree with this reviewer that it is hard to perfectly cover the surface by 50 nm KF layer due to the undulation of LLZTO ceramic surface in nm-scale although the surface has been polished carefully. The XPS data were collected from the LLZTO pellet surface with/without KF layer by thermally evaporating before cycling. As mentioned in **Question 1**, using the same thermally evaporating parameter, a dense and uniform KF layer on Si-substrate was observed clearly, and the thickness is about 50 nm (**Figure R2**). Besides, cross-sectional SEM image and EDS mappings of LLZTO-KF were shown in Figure 1c. The smooth surface and KF modifying layer were observed clearly. In addition, the AFM topography image of LLZTO-KF indicated the dense and uniform KF layer on LLZTO substrate, several holes were observed in this dense layer because of the grain boundary and voids of LLZTO substrate (**Figure R4**). Therefore, a weak Li 1s peak was collected in XPS in Figure 2 in Main Text. While, *a LiF/KF dense layer could cover the LLZTO surface completely after the reaction of KF and Li metal by heating*. The detailed experiments and discussion are given as follow.

In addition, we thermally evaporate ~100 nm lithium metal on the substrate, and then thermally evaporate a ~50 nm KF layer on the Li metal surface, as shown in Figure R5a inset. Before etching, only K-F signal was clearly collected on the top layer (KF layer) surface of F 1s regions. And classical K-F bond was observed in K 2p regions, while no markedly Li signal was detected on the sample surface. With the etching depth increasing, *the signal of generated LiF ($\text{Li}+\text{KF} \rightarrow \text{LiF}+\text{K}$) began to appear* and gradually increased according to the XPS spectrum of F 1s and Li 1s. Moreover, the K-O signal was collected at 50 nm depth and it became stronger at 100 nm (Figure R5b), where the K-O signal come from the generated K metal (K is active metal, always showing K-O signal in XPS test). For Li 1s regions, Li metal signal began to appear when etching depth arriving Li metal layer (50 nm) and lithium metal peak occupied a higher proportion when arriving the interior lithium layer (Figure R5c).

Figure R5. XPS spectra of Substrate|Li|KF samples. (a-c) XPS spectrum of F 1s, K 2p and Li 1s regions with various etching depth of 0, 20, 50 and 100 nm, respectively.

Furthermore, we thermally evaporate ~100 nm lithium metal on LZTO-KF pellet, the KF layer was sandwiched between LLZTO pellet and lithium metal as shown in inset of Figure R6a. Then put a piece of LLZTO-KF|Li pellet on hotplate, 200 °C heat treatment for 10 min. Finally, we were surprised to find that the gray lithium metal on the surface nearly disappeared after heat treatment. Figure R6a exhibited the photos of LLZTO-KF|Li pellet with/without heat treatment. XPS test was performed to analyse the surface chemical constituents. As shown in Figure R6b-d, before heat treatment, the Li_2O and Li_2CO_3

signals were obtained from the thermally evaporated lithium metal surface. Weak adsorbent F 1s peak (adsorbent F) was collected, but no any KF was found according to XPS spectrum of F 1s and K 2p regions. As the contrast, the gray lithium nearly disappeared after heat treatment. In addition to the unreacted Li metal and KF, the generated LiF and K were also observed on the sample surface. No La and Zr signals from the LLZTO pellet underneath could be collected (Figure R6e-f), indicating the dense interface layer covered the LLZTO pellet completely after the reaction (heat treatment) between KF and Li.

Figure R6. (a) The photos of LLZTO-KF|Li pellet with/without heat treatment. (b-f) XPS spectrum of Li 1s, F 1s, K 2p, La 3d and Zr 3d regions of LLZTO-KF|Li pellet with/without heat treatment, respectively.

Furthermore, we refer to the two-phase diagram of LiF-Li and KF-K reported by A. S. Dworkin *et al.* [1], LiF does not dissolve into melted Li metal below 845 °C and KF does not dissolve into melted K metal below 850 °C (Figure R7). For our LLZTO-KF|Li interface, the forming LiF can prevent the further reaction between KF and Li at the interface and the forming K does not affect the insulation of the interlayer. Therefore, the ionic-conducting and electron-blocking layer consisted of LiF and KF was *in situ* generated after LLZTO-KF reacting with melted lithium metal.

Figure R7. Liquid metal-salt phase equilibria in the alkali metal-fluoride systems (LiF-Li and KF-K systems) ^[1].

Finally, the results, discussions and figures were modified in the revised Main Text in blue text according to this comment. (Page 4, 12)

“Moreover, atomic force microscope (AFM) was employed to measure the surface morphology of LLZTO-KF pellet. Figure S5a is the AFM topography image of LLZTO-KF, and the corresponding 3D topographic image is displayed in Figure S5b, indicating the dense and uniform layer. Section analysis was given in Figure S5c, showing the surface roughness of ~25 nm.”

Figure S5 in revised supplementary information. (a) AFM topography image of LLZTO-KF ($10 \times 10 \mu\text{m}$). (b) 3D AFM topographic image of (a). (c) Height profiles along the yellow line in the 2D image of (a).

“Moreover, to verify the reaction between the KF layer and Li clearly and easily, depth-profiling X-ray photoelectron spectroscopy (XPS) analysis was performed to obtain the composition information in different depth by ion etching. We firstly thermally evaporate $\sim 100 \text{ nm}$ lithium metal on the substrate, and then thermally evaporate a $\sim 50 \text{ nm}$ KF layer on the Li metal surface in glovebox, as shown in Figure 2c inset. Before etching, as shown in Figure 2c-e, classical K-F signal was clearly collected on the top layer (KF layer) surface of F 1s and K 2p regions, while no markedly Li signal was detected on the sample surface. With the etching depth increasing, the signal of generated LiF began to appear and gradually increased according to the XPS spectrum of F 1s and Li 1s. Moreover, the K-O signal was collected at 50 nm depth and it became stronger at 100 nm (Figure 2d), where the K-O signal come from the generated K metal (Notably, K is the active metal, always showing K-O signal in XPS result). For Li 1s regions, as displayed in Figure 2e, lithium metal signal began to appear when etching depth arriving Li metal layer (50 nm) and lithium metal peak occupied a higher proportion when arriving the interior lithium layer (100 nm). According to the LiF-Li and KF-K phase diagrams as shown in Figure S10⁴¹, the forming LiF does not dissolve into Li and forming K does not dissolve into KF. For our LLZTO-KF|Li interface, the forming LiF can prevent the further reaction between KF and Li at the interface, and the forming K stay away the KF/LiF interlayer and does not affect the insulation of the interlayer. Combining the

characterization results of XRD, GI-XRD and depth-profiling XPS, we propose that the reaction mechanism is **Reaction 1**:

Specially, we find more intuitive evidence that the remarkable reaction evolution could be observed directly with the naked eyes. We thermally evaporate ~100 nm lithium metal on LZTO-KF pellet, the surface is covered by gray lithium metal. Then put a piece of LLZTO-KF|Li pellet on hotplate for 200 °C heat treatment for 10 min to accelerate the reaction process. Finally, gray lithium metal on the surface nearly disappeared after heat treatment. Figure S9a exhibited the photo of LLZTO-KF|Li pellet with/without heat treatment. XPS test was performed to analyze the surface chemical constituents. As shown in Figure S9b-d, before heat treatment, the Li₂O and Li₂CO₃ signals were obtained from the thermally evaporated lithium metal surface. Weak adsorbent F 1s peak was collected, but no any KF was found according to XPS spectrum of F 1s and K 2p regions. As the contrast, the gray lithium nearly disappeared after heat treatment. Except as the unreacted Li metal and KF, the generated LiF and K were also observed on the sample surface. No Zr and La signals were collected, indicating that the LLZTO substrate was covered by the generated dense layer completely (Figure S9e-f). This finding proves that heat treatment could promote **Reaction 1**.”

Figure 2c-e in revised Main Text. Reaction mechanism of Li and KF buffer layer. (c-e) XPS spectrum of F 1s (c), K 2p (d) and Li 1s (e) regions with various etching depth of 0, 20, 50 and 100 nm, respectively. The inset in (c) is the structure schematic diagram of Substrate|Li|KF samples.

Figure S9 in revised supplementary information. (a) The photos of LLZTO-KF|Li pellet with/without heat treatment. (b-f) XPS spectrum of Li 1s, F 1s, K 2p, La 3d and Zr 3d regions of LLZTO-KF|Li pellet with/without heat treatment, respectively.

Figure S10 in revised supplementary information. Liquid metal-salt phase equilibria in the alkali metal-fluoride systems (LiF-Li and KF-K systems). ^[1]

Question 3: Cross-sectional SEM images are provided only after cycling. Specific cycling information, such as current density and the number of cycles, is also missing. To demonstrate the behavior of the KF layer, images taken before cycling should be provided and compared with those obtained after cycling.

Response: Thanks for point it out. The provided cross-sectional SEM images in Main Text were the samples after CCD test as shown in Figure R8. Li|KF-LLZTO-KF|Li symmetrical cells displayed a stable Li-LLZTO interface, while the cells with bare LLZTO exhibited a poor Li|LLZTO interface. As shown in Figure R9, Li|LLZTO|Li cells were short circuit at 0.3 mA cm^{-2} , and then the value of potential changes dramatically, showing a serious interfacial deterioration process. While, the symmetrical cells with LLZTO-KF showed a high CCD value and soft potential change, indicating the good interface contact and stable interface.

Figure R8. Morphological characterization of the interface of Li anode and LLZTO SSEs after CCD test. Cross-sectional view SEM images of (a-b) LLZTO-KF|Li and (c-d) LLZTO|Li.

Figure R9. CCD profiles at time-constant mode of of Li|LLZTO|Li and Li|KF-LLZTO-KF|Li cells.

In addition, we have added the cross-sectional SEM images of cells before cycling and after cycling at 0.5 mA cm^{-2} for $0.5\text{h}@20$ cycles according to this comment. Figure R10 showed the SEM images before and after cycles.

Figure R10. Cross-sectional SEM images of cells with LLZTO (a, c) and LLZTO-KF (b, d) before cycling and after cycling at 0.5 mA cm^{-2} for $0.5\text{h}@20$ cycles

Finally, the results, discussions and figures were modified in the revised manuscript in blue text according to this comment. (Page 8)

“Before Li plating and stripping process, both of LLZTO|Li and LLZTO-KF|Li interfaces are tight as shown in Figure S11.”

Figure S11 in revised supplementary information. Cross-section SEM images of LLZTO|Li and LLZTO-KF|Li interfaces before cycling.

Question 4: The role of the KF layer is suggested to prevent the formation of Li dendrites, but what we observe is that void formation, rather than dendrites, is mitigated. More explanation is needed.

Response: Thanks for this comment. It is generally known that interfacial voids could produce heterogeneous local current concentration, which induces promote the growth of lithium dendrites [2-5]. When the lithiophilic and electron-blocking interface layer is added, the metal lithium deposition become uniform and flat. In this way, the interface voids and the pulverization of metal lithium will be reduced, and the growth of lithium dendrites will be restrained. As the result, the LLZTO solid state electrolytes show longer cycle life and higher current density. The flatter interface of LLZO-KF|Li was observed from SEM images due to the uniform plating and stripping of lithium metal from the interface layer. As we discussed above in **Question 2**, the KF interface layer reacted with molten lithium metal and formed LiF and K (Figure R5 and R6), which results in good Li wettability and an intimate contact between LLZO and Li metal. Therefore, the void at interfaces were prevented, and subsequently lithium plated/stripped

uniformly and the formation of Li dendrite were suppressed. For cells with bare LLZTO, the lithium dendrites and deposits were observed at the LLZTO|Li interface (Figure R11a, b) and inside LLZTO pellet (Figure R11c, d) after cycles.

Figure R11. (a, b) Cross-sectional SEM images of LLZTO|Li interface (a) and the enlarged images (b) after cycles. (c, d) SEM images of lithium dendrite in different areas inside LLZTO after cycles.

Finally, the results, discussions and figures were modified in the revised Main Text in blue text according to this comment. (Page 7)

“For further prove interface stability, cross-sectional SEM images and EDS mapping were compared of Li|SSE|Li symmetric cells with LLZTO and LLZTO-KF. Before Li plating and stripping process, both of LLZTO|Li interface and LLZTO-KF|Li interfaces are tight as shown in Figure S11. After Li plating and stripping cycles, as shown in Figure 34a-b, a tight and stable interface between lithium anode and LLZTO-KF pellet was detected. By contract, the lithium metal and LLZTO cracked severely and formed many voids in LLZTO|Li interface in Figure 3c-d, where the voids can cause heterogeneous current density distribution and induce the formation of lithium dendrites. Moreover, the lithium dendrites and deposits

were observed at the LLZTO|Li interface (Figure S12a-b) and inside LLZTO pellet (Figure S12c-d) after cycles. When the lithiophilic and electron-blocking interface layer is added, the metal lithium deposition become uniform and flat. In this way, the interface voids and the pulverization of lithium metal anode will be reduced, and the growth of lithium dendrites will be restrained.”

Figure S11 in revised supplementary information. Cross-section SEM images of LLZTO|Li and LLZTO-KF|Li interfaces before cycling.

Figure S12 in revised supplementary information. (a, b) Cross-sectional SEM images of LLZTO|Li interface (a) and the enlarged images (b) after cycles. (c, d) SEM images of lithium dendrite in different areas inside LLZTO after cycles.

Question 5: The COMSOL simulation does not seem relevant to the core concept of this work. Physical and transport properties of KF are not reflected in the simulation. It only simulates the influence of voids at the interface, which is highly predictable. I do not believe this simulation study provides valuable insight or demonstrates anything meaningful.

Response: Thanks for pointing it out. We have modified the new models to simulate the current density distribution at the interface according to this comment. In the modified $4 \times 3 \mu\text{m}^2$ 2D geometry models (Figure 3f-g), the top area was LLZTO with or without electron-blocking and lithiophilic layer, the bottom area was the Li anode. The electron blocking and lithiophilic layer was added, which reduce the interface resistance and improve the ionic transfer uniformly, resulting the homogeneous interface reaction and current density distribution. Meanwhile, the control model without modifying layer cause heterogeneous reaction and heterogeneous current density distribution at the interface.

Finally, the results, discussions and figures were modified in the revised Main Text in blue text according to this comment. (Page 8 and 13)

Figure 3f-g in revised Main Text. The finite element analysis results of current density distribution in LLZO-KF|Li interface (f) and LLZO|Li interface (g).

“The finite element analysis based on COMSOL software was employed to further verify the phenomena. As shown in Figure 3f, the tight LLZTO-KF|Li interface displayed a homogeneous current density distribution thanks to the KF/LiF interlayer. Correspondingly, LLZTO|Li interface with voids shows many local current density hotspots (Figure 3g), in which heterogeneous local current density distribution at the interface may trigger the preferential growth of lithium metal dendrite.”

“The simulation results were obtained from COMSOL software with “Lithium-Ion Battery interface”. A simplified cell model was used to simulate the electric field distribution at the interface between Li anode and LLZTO solid state electrolyte with COMSOL Multiphysics software. In the $4 \times 3 \mu\text{m}^2$ 2D geometry models (Figure S27), the top area was LLZTO with or without electron-blocking and lithiophilic layer, the bottom area was the Li anode. The electron blocking and lithiophilic layer could improve the interface reaction uniformly. Meanwhile, the control model without buffer layer cause heterogeneous reaction at the interface.”

Figure S27 in revised supplementary information. Simulation cell geometry in COMSOL for LLZO-KF|Li interface (a) and LLZO|Li interface (b).

Question 6: The Li ion conductivity of the KF layer should be provided or measured. Additionally, the added resistance caused by the 50 nm thick KF layer should be calculated, and its impact on overall cell performance should be discussed.

Response: Thanks for this suggestion. We have measured the EIS data of the LLZTO pellets with/without KF layer at room temperature. Figure R12 displayed the Nyquist plots of several LLZTO and LLZTO-KF pellets. The average ionic conductivity of LLZTO and LLZTO-KF were calculated both as $\sim 4.2 \times 10^{-4} \text{ S cm}^{-1}$, indicating that the ionic conductivity of the samples was not affected by KF layer.

Figure R12. (a, b) Nyquist plots of LLZTO and LLZTO-KF pellets at room temperature.

Moreover, we have measured the EIS of LLZTO and LLZTO-KF samples at various temperature and calculated the activation energy. Figure R13 exhibited the Nyquist plots and Arrhenius plots of the

samples ranged from 0 to 80°C. The activation energy of LLZTO and LLZTO-KF is calculated as 0.40 eV and 0.39 eV, respectively.

Figure R13. Nyquist plots and Arrhenius plots of Au|LLZTO|Au and Au|KF-LLZTO-KF|Au symmetric cells at various temperatures. (a, b) Nyquist plots and enlarged plots of Au|LLZTO|Au cells. (c) Arrhenius plots of Au|LLZTO|Au symmetric cells. (d, e) Nyquist plots and enlarged plots of Au|KF-LLZTO-KF|Au cells. (f) Arrhenius plots of Au|KF-LLZTO-KF|Au symmetric cells.

In addition, we also performed DFT calculations to compare the Li-ion migration barriers in the KF interlayer, LiF interlayer and Li₂CO₃ interlayer on the LLZTO surface (Li₂CO₃ is easily generated on the surface of bare LLZTO). **Figure 3h-k** showed that the migration barrier of Li-ion in KF and LiF interlayer are 0.80 eV and 1.08 eV, which are significantly lower than that of Li₂CO₃ interlayer of 2.62 eV.

Finally, the results, discussions and figures were modified in the revised Main Text in blue text according to this comment. (Page 8)

“Furthermore, density functional theory (DFT) calculations were performed to attain deep insight in interface chemistry. The interfacial formation energy of KF|LLZTO was calculated as -2.28 J m^{-2} (see methods for details), which represent the good interface wettability between LLZTO and KF. Li diffusion energy with the lowest barrier was shown in Figure 3h. According to previous report, it is easy for bare

LLZTO surface to form a lithiophobic Li_2CO_3 layer with low lithium-ion conductivity in the air^{22,27-29}. And $\text{Li}_2\text{CO}_3|\text{LLZTO}$ (Figure 3i) showed a quite high diffusion barrier of 2.62 eV, which may cause the lithium accumulation in the local region and then bring about the formation of lithium dendrites. On the contrary, the migration energy barriers of Li^+ at $\text{KF}|\text{LLZTO}$ and $\text{LiF}|\text{LLZTO}$ (Figure 3j-k) is calculated as 0.80 eV and 1.08 eV, respectively, which are lower than that of $\text{Li}_2\text{CO}_3|\text{LLZTO}$ (2.62 eV). The low migration energy barriers indicate fast ion transport across the interface, which further facilitate the uniform distribution of Li^+ , and effectively inhibit the formation of Li dendrites.

Figure 3h-k in revised Main Text. (h), Li diffusion energy with the lowest barrier. (i-k) The migration path in Li_2CO_3 (i), $\text{LLZTO}|\text{KF}$ (j) and $\text{LLZTO}|\text{LiF}$ (k), the Li migration path is denoted as yellow balls.

“Next, the electrochemical performances of LLZTO and LLZTO-KF were measured. The EIS of LLZTO and LLZTO-KF samples were measured at various temperature. Figure S13 exhibited the Nyquist plots and Arrhenius plots of the samples ranged from 0 to 80°C. Arrhenius plots of $\text{Au}|\text{LLZTO}|\text{Au}$ and $\text{Au}|\text{KF-LLZTO-KF}|\text{Au}$ symmetric cells were shown in Figure 5a, and the activation energy of LLZTO and LLZTO-KF is calculated as 0.40 eV and 0.39 eV, respectively, according to the equation 1.

$$\sigma = A \exp\left(-\frac{E_a}{kT}\right) \quad (\text{Equation 1})$$

Where σ is ionic conductivity at different temperature, T is thermodynamic temperature, A is pre-exponential factor, E_a is activation energy, and k is Boltzmann constant.”

Figure S13 in revised supplementary information. Nyquist plots and Arrhenius plots of Au|LLZTO|Au and Au|KF-LLZTO-KF|Au symmetric cells at various temperatures. (a, b) Nyquist plots and enlarged plots of Au|LLZTO|Au cells. (c, d) Nyquist plots and enlarged plots of Au|KF-LLZTO-KF|Au cells.

Figure 5a in revised Main Text. The Arrhenius plots of the LLZTO-KF and LLZTO.

Question 7: LiTFSI in C₄mim is chosen as the catholyte, but its performance is not adequate for use in commercial cells. The cathode loading is approximately 2.5 mg/cm², corresponding to an areal capacity of 0.5 mAh/cm² or lower. Therefore, 0.5 C would be 0.25 mA/cm² or lower. The fact that only 80% of the capacity is obtained at such low current density and low loading suggests that this catholyte is not suitable. In typical Li-ion batteries, the areal capacity is higher than 3 mAh/cm², and more than 90% of capacity is recovered at 0.5 C (compared to 0.1 C capacity).

Response:

This specific capacity performance was measured in the solid-state Li|KF-LLZTO-IL|NCM cells, which is the different battery system from commercial liquid batteries. It is not easy to compare solid state batteries and commercial liquid lithium-ion batteries. We summarized the electrochemical performances including coulombic efficiencies and cycle life at various C-rates of related solid-state Li metal batteries in Fig. 6e, *indicating that the electrochemical performance of our Li|KF-LLZTO-IL|NCM cells is better than that of most of the literatures*. In addition, the advantage of ionic liquid is not only the areal specific capacity, but also the replacement of liquid flammable organic electrolyte to improve safety. According to this comment, we assembled the Li|KF-LLZTO-IL|NCM cells with **higher mass loading of ~4 mg cm⁻²** NCM cathode and cycled at 0.2C. As shown in *Figure S23*, the cells have an initial specific capacity of 149.9 mAh g⁻¹, with the capacity retention of 83.8% after 125 cycles. And we will focus on the high mass loading cathode in our further work.

Figure 6e in revised Main Text. Cycle performance comparison chart of our work with other literature reports.

Finally, the results, discussions and figures were modified in the revised Main Text in blue text according to this comment. (Page 10)

“Also, high mass loading NCM of $\sim 4 \text{ mg cm}^{-2}$ was adopted and cycled at 0.2C. As displayed in Figure S23, the cells with LLZTO-KF have a high initial specific capacity of 145.9 mAh g^{-1} , with a capacity retention of 83.8% after 125 cycles.”

Figure S23 in revised supplementary information. Long cycle performance of Li|KF-LLZTO-IL|NCM cells at 0.2C.

Question 8: There are numerous grammatical errors and incorrect word usage in the manuscript. Here are a few examples, but there are many more.

"Garnet oxide" "physicochemical (.....)" "security risks" "In order to comparing" "for fully reaction" "after nature cooling" "the Extreme electron-blocking" "were compared of"

Response: Thanks for this comment. We have checked this manuscript carefully and modified the corresponding descriptions and discussions.

Reference

- [1] Dworkin, A. S., H. R. Bronstein, and M. A. Bredig. Miscibility of metals with salts. vi. lithium-lithium halide systems1. *The Journal of Physical Chemistry* 66.3 (1962): 572-573.
- [2] Lee, Sunyoung, et al. Design of a lithiophilic and electron-blocking interlayer for dendrite-free lithium-metal solid-state batteries. *Science Advances* 8.30 (2022): eabq0153.

- [3] Huo, Hanyu, et al. A flexible electron-blocking interfacial shield for dendrite-free solid lithium metal batteries. *Nature Communications* 12.1 (2021): 176.
- [4] Bi, Zhijie, et al. Molten Salt Driven Conversion Reaction Enabling Lithiophilic and Air - Stable Garnet Surface for Solid - State Lithium Batteries. *Advanced Functional Materials* 32.52 (2022): 2208751.
- [5] Chen, Butian, et al. Constructing a Superlithiophilic 3D Burr - Microsphere Interface on Garnet for High-Rate and Ultra-Stable Solid-State Li Batteries. *Advanced Science* 10.11 (2023): 2207056.

Reviewer #3 (Remarks to the Author):

Authors demonstrate that a highly effective interlayer composed of potassium fluoride (KF) can inhibit lithium dendrite growth in garnet oxide (LLZTO), resulting in an ultralong cycle life of over 3000 hours at 0.2 mA cm⁻² and over 350 hours at 0.5 mA cm⁻² in room temperature. The paper also discusses the mechanism behind the KF interlayer's ability to block electro migration and inhibit dendrite growth, as well as the potential applications of this approach in the development of solid-state lithium-metal batteries. This paper presents a novel approach to developing an extreme electron-blocking interface for garnet-based ASSBs. The manuscript is well written systematically; however, I found some few doubts. After addressing, it will be published in high-quality journal.

Response: We would like to appreciate this reviewer's positive comments and valuable suggestion. We have designed corresponding experiment, modified some descriptions and details, and added more discussions in the revised manuscript.

Question 1. Were the electrochemical impedances measured not only in full-cell configurations but also in electron-blocking and ion-blocking cell configurations at various temperatures, ranging from room temperature to 65°C.

Response: Thanks for review's comment. We have measured the EIS data of Au|LLZTO|Au and Au|KF-LLZTO-KF|Au symmetric cells at various temperatures. Figure R1 exhibited the Nyquist plots and Arrhenius plots of the samples. The activation energy of LLZTO and LLZTO-KF is calculated as 0.40 eV and 0.39 eV, respectively according to the equation 1.

$$\sigma = A \exp\left(-\frac{E_a}{kT}\right) \quad (\text{Equation 1})$$

Where σ is ionic conductivity at different temperature, T is thermodynamic temperature, A is pre-exponential factor, E_a is activation energy, and k is Boltzmann constant.

Figure R1. Nyquist plots and Arrhenius plots of Au|LLZTO|Au and Au|KF-LLZTO-KF|Au symmetric cells at various temperatures. (a, b) Nyquist plots and enlarged plots of Au|LLZTO|Au cells. (c) Arrhenius plots of Au|LLZTO|Au symmetric cells. (d, e) Nyquist plots and enlarged plots of Au|KF-LLZTO-KF|Au cells. (f) Arrhenius plots of Au|KF-LLZTO-KF|Au symmetric cells.

And the results, discussions and figures were modified in the revised manuscript in blue text according to this comment. (Page 8)

“Next, the electrochemical performances of LLZTO and LLZTO-KF were measured. The EIS of LLZTO and LLZTO-KF samples were measured at various temperature. Figure S13 exhibited the Nyquist plots and Arrhenius plots of the samples ranged from 0 to 80°C. Arrhenius plots of Au|LLZTO|Au and Au|KF-LLZTO-KF|Au symmetric cells were shown in Figure 5a, and the activation energy of LLZTO and LLZTO-KF is calculated as 0.40 eV and 0.39 eV, respectively, according to the equation 1.

$$\sigma = A \exp\left(-\frac{E_a}{kT}\right) \quad (\text{Equation 1})$$

Where σ is ionic conductivity at different temperature, T is thermodynamic temperature, A is pre-exponential factor, E_a is activation energy, and k is Boltzmann constant.”

Figure S13 in revised supplementary information. Nyquist plots and Arrhenius plots of Au|LLZTO|Au and Au|KF-LLZTO-KF|Au symmetric cells at various temperatures. (a, b) Nyquist plots and enlarged plots of Au|LLZTO|Au cells. (c, d) Nyquist plots and enlarged plots of Au|KF-LLZTO-KF|Au cells.

Figure 5a in revised Main Text. The Arrhenius plots of the LLZTO-KF and LLZTO.

Question 2. The crystallinity of the KF phase is ambiguous. The XRD peaks matched with PDF#36-1458, but Figure 3a showed PDF#85-1314 after Li-KF reactions. Does the deposited KF on LLZTO exhibit crystallinity and remain stable after multiple cycles?

Response: Thanks for point it out. We apologized that we confused the two standard cards uncarefully. PDF#85-1314 standard cards should be used in both Main Text and Figure 3a.

In addition, synchrotron radiation GI-XRD was employed to affirm the phase structure of LLZTO-KF (~50 nm KF layer on LLZTO pellet). Figure R2a-b displayed the synchrotron radiation GI-XRD patterns and enlarged images of LLZTO-KF pellet with the incidence angle of 0.05, 0.1, 0.2, 0.5 and 1°. No KF peaks were observed, indicating that the initial KF layer may be not crystal phase. Interestingly, after reacting with melted lithium metal, the peaks of Li, KF and LiF were collected from LLZTO-KF|Li sample and matched well with PDF cards (Figure R2c-d). The results also prove that the reaction mechanism **Reaction 1**. (See more details in Question 5).

According to the two-phase diagram of LiF-Li and KF-K reported by A. S. Dworkin *et al.* ^[1], LiF does not dissolve into melted Li metal below 845 °C and KF does not dissolve into melted K metal below 850 °C (Figure S10). For our LLZTO-KF|Li interface, the forming LiF can prevent the further reaction between KF and Li at the interface and the forming K does not affect the insulation of the interlayer. Therefore, the ionic-conducting and electron-blocking layer consisted of LiF and KF was *in situ* generated after LLZTO-KF reacting with melted lithium metal.

Figure R2. Synchrotron radiation GI-XRD patterns of LLZTO-KF pellet and LLZTO-KF|Li pellet under Kapton film. (a, b) Synchrotron radiation GI-XRD patterns and enlarged images of LLZTO-KF pellet at various incidence angle. (c, d) Synchrotron radiation GI-XRD patterns and enlarged images of LLZTO-KF|Li pellet at various incidence angle.

Figure S10 in revised supplementary information. Liquid metal-salt phase equilibria in the alkali metal-fluoride systems (LiF-Li and KF-K systems) ^[1].

Moreover, Figure 3a-e in revised Main Text exhibited the cycle stability of LLZTO-KF|Li interface. As shown in Figure 3a-b, a tight and stable interface between lithium anode and LLZTO-KF pellet was detected after cycles. By contrast, the lithium metal and LLZTO cracked severely in LLZTO|Li interface in Figure 3c-d. Figure 3e is the corresponding EDS mapping images of LLZTO-KF|Li interface after cycles. F element at KF interlayer was observed visibly, representing the good wettability and stability. The stable interface could be attributed to the KF/LiF electron blocking and ion conducting layer, which forbade lithium dendrite growth and reduce local current density at interface.

Figure 3a-e in revised Main Text. Morphological characterization and EDS mapping images of the interface of Li anode and LLZTO SSEs after cycles. Cross-sectional view SEM images of (a-b) LLZTO-KF and (c-d) LLZTO. (e), The corresponding EDS mapping images of (b).

Finally, the results, discussions and figures were modified in the revised manuscript in blue text according to this comment. (Page 5, 7 and 8)

In addition, the phase structure of LLZTO-KF/Li sample (dipping LLZTO-KF pellet into melted lithium fleetly) was verified by synchrotron radiation GI-XRD with different incidence angle of 0.2° , 0.5° and 1° (Figure S8 and Figure 2b). Thanks to the high energy and high resolution of the synchrotron radiation X-ray, the phase structure of interlayer with very low content has been successfully detected. Figure 2b displayed the enlarged synchrotron radiation GI-XRD pattern at 1° , except the LLZTO substrate, the peaks of KF (28.9° , 33.5° and 48.1°), LiF (38.7° and 45°) and Li (36.2°) were observed clearly.

Figure 2b in revised Main Text. Synchrotron radiation GI-XRD patterns of LLZTO-KF|Li sample at the incidence angle of 1°.

Figure S8 in revised supplementary information. Synchrotron radiation GI-XRD patterns (a) and the enlarged picture (b) of LLZTO-KF|Li sample at the incidence angle of 0.2°, 0.5° and 1°.

According to the LiF-Li and KF-K phase diagrams as shown in Figure S10⁴¹, the forming LiF does not dissolve into Li and forming K does not dissolve into KF. For our LLZTO-KF|Li interface, during assembling LLZTO-KF with Li metal anode, KF reacted with molten Li metal partially, and the forming LiF could prevent the further reaction between KF and Li at the interface. And, the forming K with very low melting pointing might diffuse into molten Li metal during reaction, which did not affect the insulation of the interlayer. Therefore, the results of XRD, GI-XRD and depth-profiling XPS supported the above reaction mechanism. The in-situ reaction of KF and Li metal improves the lithiophilic ability and reduces the interface impedance, resulting in excellent electrochemical performance.

“For further prove interface stability, cross-sectional SEM images and EDS mapping were compared between Li|SSE|Li symmetric cells with LLZTO and LLZTO-KF. Before Li plating and stripping process, both of LLZTO|Li interface and LLZTO-KF|Li interfaces are tight as shown in Figure S11. After Li plating and stripping cycles, as shown in Figure 3a-b, a tight and stable interface between lithium anode and LLZTO-KF pellet was detected. By contract, the lithium metal and LLZTO cracked severely and formed many voids in LLZTO|Li interface in Figure 3c-d, where the voids can cause heterogeneous current density distribution and induce the formation of lithium dendrites. Moreover, the lithium dendrites and deposits were observed at the LLZTO|Li interface (Figure S12a-b) and inside LLZTO pellet (Figure S12c-d) after cycles. When the lithiophilic and electron-blocking interface layer is added, the metallic lithium deposition become uniform and flat. In this way, the interface voids and the pulverization of lithium metal anode reduced, and the growth of lithium dendrites restrained. Figure 3e is the corresponding EDS mapping images of LLZTO-KF|Li interface. F element at KF/LiF interlayer and Ta element at LLZTO area were observed visibly, representing the good wettability and stability. The stable interface could be attributed to the KF/LiF electron blocking layer, which forbade lithium dendrite growth and reduce local current density at interface.”

Question 3. The COMSOL simulation displays the electrical current distribution between electrolytes and electrodes. However, technically, electrical current density cannot flow through the electrolyte phase. Can the geometrical differences due to voids support the effect of KF on LLZTO? The focusing of current density near the narrow interface seems to result from geometrical effects, not the KF interface. More comprehensive details are required for clarity.

Response: Thanks for pointing it out. We have modified the new models to simulate the current density distribution at the interface according to this comment. In the modified $4 \times 3 \mu\text{m}^2$ 2D geometry models (*Figure 3f-g*), the top area was LLZTO with or without electron-blocking and lithiophilic layer, the bottom area was the Li anode. The electron blocking and lithiophilic layer was added, which reduce the interface resistance and improve the ionic transfer uniformly, resulting the homogeneous interface reaction and current density distribution. Meanwhile, the control model without buffer layer cause heterogeneous reaction and heterogeneous current density distribution at the interface.

Finally, the results, discussions and figures were modified in the revised Main Text in blue text according to this comment. (Page 8 and 13)

“The finite element analysis based on COMSOL software was employed to further verify the phenomena. As shown in Figure 3f, the tight LLZTO-KF|Li interface with modifying layer displayed a homogeneous current density distribution thanks to the KF/LiF interlayer. Correspondingly, LLZTO|Li interface with voids shows many local current density hotspots (Figure 3g), in which heterogeneous local current density distribution at the interface may trigger the preferential growth of lithium metal dendrite.”

Figure 3f-g in revised Main Text. The finite element analysis results of current density distribution in LLZO-KF|Li interface (f) and LLZO|Li interface (g).

“ The simulation results were obtained from COMSOL software with “Lithium-Ion Battery interface”. A simplified cell model was used to simulate the electric field distribution at the interface between Li anode and LLZTO solid state electrolyte with COMSOL Multiphysics software. In the $4 \times 3 \mu\text{m}^2$ 2D geometry models (Figure S27), the top area was LLZTO with or without electron-blocking and lithiophilic layer, the bottom area was the Li anode. The electron blocking and lithiophilic layer could improve the interface reaction uniformly. Meanwhile, the control model without buffer layer cause heterogeneous reaction at the interface.”

Figure S27 in revised supplementary information. Simulation cell geometry in COMSOL for LLZO-KF|Li interface (a) and LLZO|Li interface (b).

Question 4. In reference 47, Huo et al. in Nat Commun (2021) proposed a similar concept of an electron-blocking layer between LLZTO/Li. To validate the role of electron-blocking by PAA, they demonstrated depth profiles of TOF-SIMS for LLZTO/PAA pellets and performed electronic calculations using DFT simulations. In this study, although the authors suggested the structure of KF as PDF#85-1314, the details of the structure and the origin of electron-blocking by KF itself are not clear.

Response: Thanks for this comment. According to this suggestion, we have added TOF-SIMS and DFT calculations, modified some descriptions and details, and added more discussions in the revised Main Text.

According to this suggestion, time-of-flight secondary-ion mass spectroscopy (TOF-SIMS) was employed to further verify the KF layer. Figure R3a showed the depth profiles of KF on the surface of LLZTO pellet, which reveals the evolution of several fragments as the sputtering proceeds in a negative mode. K^- and KF_2^- come from KF layer on the surface of LLZTO-KF, and TaO_2^- and ZrO_2^- signals represent LLZTO ceramic pellets (substrate). Figure R3b is the 3D view of element distribution, and Figure R3c shows the element distribution map in 2D view of y-z plane. The K^- and KF_2^- signals are strong in the beginning, and then gradually declined with the sputtering time (etching depth). On the contrary, the TaO_2^- and ZrO_2^- fragments signal is very weak when it appears, increases almost in parallel with the sputtering time, and finally tends to be stable, which means that the LLZTO-KF surface was covered with a dense and homogeneous KF interlayer. Therefore, the existence of KF on the LLZTO-KF surface is proved, which is consistent with the results of XPS.

Figure R3. (a) TOF-SIMS depth profiles for the LLZTO-KF pellet. (b) 3D view of element distribution. (c) 2D view of element distribution of y-z planes.

Moreover, we use TOF-SIMS to further explore the interlayer evolution of LLZTO-KF|Li surface. As shown in Figure R4a inset, the top layer of the sample is ~ 200 nm lithium metal layer by thermally evaporating, the bottom layer is LLZTO pellet, and the KF layer is sandwiched between them. Figure R4a displays the TOF-SIMS depth profiles of the LLZTO-KF|Li sample and the enlarged profiles is shown in Figure R4b. Firstly, remarkable LiO^- and K^- signal are collected from the top lithium layer, where LiO^- comes from lithium metal layer. According to the LiF-Li and KF-K phase diagrams as shown in Figure S10 in revised supplementary information, the forming LiF does not dissolve into Li and forming K does not dissolve into KF . Therefore, K^- may represent the forming K that might diffuse into molten Li metal during reaction ($\text{Li} + \text{KF} \rightarrow \text{LiF} + \text{K}$, see more details in Question 5). In addition, the LiF_2^- and K^- fragments from middle layer indicate the KF/LiF mix layer, which blocks the electron transport but conducts Li-ion . The modifying interlayer could improve the lithium wettability and inhibit the

formation of lithium dendrites. With the sputtering proceeding, strong TaO_2^- , LaO_2^- and ZrO_2^- signals are observed from bottom LLZTO layer. Notably, the fluorine element diffuses easily, so the LiF_2^- signal began to be strong in the KF/LiF layer as well as the later LLZTO substrate layer.

Figure R4. (a) TOF-SIMS depth profiles for the LLZTO-KF|Li sample, the inset is the structure schematic diagram of LLZTO-KF|Li. (b) 3D view of element distribution. (c) 2D view of element distribution of y-z planes.

As we discussed above and **Question 2** as well as **Question 5**, KF react with lithium metal, and generate LiF and K. Therefore, the electron-blocking KF interlayer *in situ* transformed to an electron-blocking and Li-ion conducting LiF/KF interlayer. Therefore, DFT simulations of KF|Li interface (Figure 4a, d, g in revised Main Text), LiF|Li interface (Figure 4b, e, h in revised Main Text) and bare LLZTO|Li interface (Figure 4c, f, i in revised Main Text) were employed to verify the electron-blocking property. Figure 4a-

c are the structure and charge transfer conditions, and corresponding electrostatic potential profiles of interface are shown in Figure 4d-f. KF and LiF layer could effectively blocks electrons at the interface, which is confirmed by the electrostatic potential profiles shown in Figure 4d-e. The electrostatic potential barriers (ΔE) of -4.41 eV and -2.08 eV were obtained at KF|Li interface and LiF|Li interface, respectively. The electrostatic potential barriers indicate that electron-blocking property from Li metal to interlayer. Electrons are contained to the Li metal, and Li deposition occurs preferentially at the interface between lithium and LiF/KF layer, rather than within LLZTO or the surface of LLZTO, which prohibits the penetration of Li dendrites. The electronically insulating nature is further confirmed by DOS results for KF|Li and LiF|Li, shown in Figure 4g-h. In contrast, there is no barrier ($\Delta E=8.92$ eV >0 , Figure 4f) to the transfer of electrons from the Li metal to the bare LLZTO electrolyte. In the case of bare LLZTO|Li, electrons and Li atoms preferentially deposit within the LLZTO, and this behavior corroborated by DOS results (Figure 4i).

Figure 4 in revised Main Text. Electron-blocking property of the LLZTO-KF|Li interface. (a-c) The structure and charge transfer of KF|Li interface (a), LiF|Li interface (b) and bare LLZTO|Li interface (c). (d-f) The corresponding electrostatic potential profiles of interface. (g-i) The corresponding density of

states (DOS) of interface.

Fluorine element (F) has the largest electronegativity of $\chi=4$, and potassium element (K) shows the lowest electronegativity of $\chi=0.82$ among common metallic elements. Meanwhile, KF has a band gap of $5.95 \text{ eV} > 5 \text{ eV}$ (Forbidden band, electronic insulator). Theoretically, the ionic compound potassium fluoride (KF) has strong structural stability and electronic insulation owing to the largest electronegativity difference value of $\Delta\chi=3.18$ and wide band gap of 5.95 eV . Moreover, we prepared the KF pellet (diameter=12.7 mm, thickness=1 mm) in a stainless-steel die under a uniaxial pressure of 300 MPa, and then direct current (DC) polarization test was performed with the DC voltage of 1 V and 5 V. As shown in Figure R5, the current value is too small to be accurately detected (the polarized current value fluctuates around 0), indicating the extreme low electronic conductivity of KF.

Figure R5. DC polarization plots of KF pellet with the DC voltage of 1V and 5 V.

Finally, the results, discussions and figures were modified in the revised Main Text in blue text according to this comment. (Page 4-7)

“Time of flight secondary ion mass spectrometry (TOF-SIMS) was carried out to analyse the LLZTO-KF surface. As displayed in Figure 1d, the depth profiling of LLZTO-KF reveals the evolution of several fragments as the sputtering proceeds. K^+ and KF_2^+ come from KF layer on the surface of LLZTO-KF, and TaO_2^+ and ZrO_2^+ signals represent LLZTO ceramic pellets (substrate). Figure 1e shows the 3D view of element distribution, and Figure 1f is the element distribution map in 2D view of y-z plane. The K^+ and KF_2^+ signals are strong in the beginning, and then gradually declined with the sputtering time (etching

depth). On the contrary, the TaO_2^- and ZrO_2^- fragment signals are very weak when it appears, increases almost in parallel with the sputtering time, and finally tends to be stable, which means that the LLZTO-KF surface was covered with a homogeneous KF interlayer, consisting with the results of XPS. Combining the above characterization data, it was proved that the KF layer was successfully deposited on the LLZTO-KF surface.”

“Furthermore, DFT simulations of KF|Li interface, LiF|Li interface, and bare LLZTO|Li interface were employed to verify the electron-blocking property. Figure 4a-c are the structure and charge transfer conditions. And the corresponding electrostatic potential profiles of interface are shown in Figure 4d-f. KF layer and LiF layer could effectively block electrons at the interface, which is confirmed by the electrostatic potential profiles shown in Figure 4d-e. The electrostatic potential barriers (ΔE) of -4.41 eV and -2.08 eV were obtained at KF|Li interface and LiF|Li interface, respectively. The electrostatic potential barriers indicate that electron-blocking property from Li metal to interlayer. Electrons are contained to the Li metal, and Li deposition occurs preferentially at the interface between lithium and LiF/KF layer, rather than within LLZTO or the surface of LLZTO, which prohibits the penetration of Li dendrites. The electronically insulating nature is further confirmed by DOS results for KF|Li and LiF|Li, shown in Figure 4g-h. In contrast, there is no barrier ($\Delta E=8.92$ eV >0 , Figure 4f) to the transfer of electrons from the Li metal to the bare LLZTO electrolyte. In the case of bare LLZTO|Li interface, electrons and Li atoms preferentially deposit within the LLZTO, and this behavior also corroborated by DOS results (Figure 4i).”

Question 5. Potassium fluoride is thermodynamically less stable than lithium fluoride. If excess lithium metal grows, KF may be replaced by LiF. The thermodynamic enthalpy for the reaction of KF with Li to form LiF and K is -0.507 eV or -49.055 kJ mol⁻¹.

Response: We thank the reviewer’s valuable comment and constructive suggestions. We looked in the ChemicalAid database at internet (<https://www.chemicalaid.com/>), and found the change of Gibbs free energy (ΔG) of the reaction equation, in which the ΔG of reaction 2 is approximate to reviewer’s data.

In real experiment, the KF layer will contact with melted lithium metal, so the corresponding reaction

should be reaction 3, KF layer reacted with melted lithium metal, forming solid LiF and liquid K (the melting point of Li and K are ~ 180 and ~ 64 ° C, respectively). And the detailed experiments and supporting evidences are as follows.

For our solid-state lithium metal cells, the lithium anode was prepared by dipping the LLZTO-KF pellet into molten lithium, where lithium metal was tightly bonded to pellet and it was difficult to separate them apart to observe the phase change of the interlayer material. In order to observe the reaction product of Li and KF, we have designed two kinds of experiments as follow: (1) KF powder react with melted Li metal, and (2) KF green pellet react with melted Li metal.

(1), put some KF powder into melted Li metal on stainless steel substrate by 300 °C hotplate for fully reaction.

(2), make a KF pellet in a stainless-steel die ($d=12.7$ mm) under a uniaxial pressure of 300 MPa, and put the KF pellet into melted Li metal for 30 min. Then take out the KF pellet from melted Li (KF surface was covered by Li metal), and put on the 300 °C hotplate for fully reaction.

Then, XRD and grazing incidence XRD (GI-XRD) were carried out with the two samples. For sample 1# (KF powder into melted Li metal), strong XRD peaks of Li (PDF#15-0401) and KF (PDF#85-1314) were collected as shown in Figure R6a (black line). More importantly, the weak XRD peaks of reaction product LiF (PDF#45-1460) were observed successfully. For a clearer detection of the reaction products, as suggested by this reviewer, GI-XRD was performed with an incidence angle of 1° . And stronger peaks of LiF were observed clearly (Figure R6b, black line). For sample 2# (melted Li on KF pellet), strong and clear peaks of *Li*, *KF* and *LiF* were detected from XRD and GI-XRD patterns (Figure R6ab, blue lines).

Figure R6. The XRD (a) and GI-XRD (b) patterns with an incidence angle of 1° of Li-KF reaction product

under Kapton film.

In addition, the phase structure of LLZTO-KF|Li sample (dipping LLZTO-KF pellet into melted lithium fleetly) was verified by synchrotron radiation GI-XRD with different incidence angle of 0.2° , 0.5° and 1° (Figure R7a). Thanks to the high energy and high resolution of the synchrotron radiation X-ray, the phase structure of interlayer with very low content has been successfully detected. Figure R7b displayed the enlarged synchrotron radiation GI-XRD patterns, except the LLZTO substrate, *the peaks of KF (28.9° , 33.5° and 48.1°), LiF (38.7° and 45°) and Li (36.2°) were observed clearly.*

Figure R7. Synchrotron radiation GI-XRD patterns (a) and the enlarged picture (b) of LLZTO-KF|Li sample at the incidence angle of 0.2° , 0.5° and 1° .

Moreover, to verify the reaction between the KF layer and Li clearly and easily, depth-profiling X-ray photoelectron spectroscopy (XPS) analysis was performed to obtain the composition information in different depth by ion etching. We firstly thermally evaporate ~ 100 nm lithium metal on the substrate, and then thermally evaporate a ~ 50 nm KF layer on the Li metal surface in glovebox, as shown in Figure R8a inset. Before etching, only K-F signal was clearly collected on the top layer (KF layer) surface of F 1s regions. And classical K-F bond was observed in K 2p regions, while no markedly Li signal was detected on the sample surface. With the etching depth increasing, the signal of generated LiF began to appear and gradually increased according to the XPS spectrum of F 1s and Li 1s. Moreover, the K-O signal was collected at 50 nm depth and it became stronger at 100 nm (Figure R8b), where the K-O signal come from the generated K metal (K is active metal, always showing K-O signal in XPS test). For Li 1s regions, as displayed in Figure R8c, lithium metal signal began to appear when etching depth arriving Li metal layer (50 nm) and lithium metal peak occupied a higher proportion when arriving the interior

lithium layer. Combining the characterization results of XRD, GI-XRD and depth-profiling XPS, we can prove that the reaction mechanism is **Reaction 1** discussed above.

Figure R8. XPS spectra of Substrate|Li|KF samples. (a-c) XPS spectrum of F 1s, K 2p and Li 1s regions with various etching depth of 0, 20, 50 and 100 nm, respectively.

Specially, we found more intuitive evidence that the marked reaction process could be observed directly with the naked eye. We thermally evaporate ~100 nm lithium metal on LZTO-KF pellet, the surface was covered by gray lithium metal. Then put a piece of LLZTO-KF|Li pellet on hotplate for 200 °C heat treatment for 10 min. Finally, we were surprised to find that the gray lithium metal on the surface nearly disappeared after heat treatment. Figure R9a exhibited the photo of LLZTO-KF|Li pellet with/without heat treatment. XPS test was performed to analyse the surface chemical constituents. As shown in Figure R9b-d, before heat treatment, the Li₂O and Li₂CO₃ signals were obtained from the thermally evaporated lithium metal surface. Weak adsorbent F 1s peak was collected, but no any KF was found according to XPS spectrum of F 1s and K 2p regions. As the contrast, the gray lithium nearly disappeared after heat treatment. In addition to the unreacted Li metal and KF, the generated LiF and K were also observed on the sample surface. This finding proves that heat treatment could promote **Reaction 1**.

Figure R9. (a) The photos of LLZTO-KF|Li pellet with/without heat treatment. (b-d) XPS spectrum of Li 1s, F 1s and K 2p regions of LLZTO-KF|Li pellet with/without heat treatment, respectively.

Furthermore, we refer to the two-phase diagram of LiF-Li and KF-K reported by A. S. Dworkin *et al.* ^[1], LiF does not dissolve into melted Li metal below 845 °C and KF does not dissolve into melted K metal below 850 °C (Figure R10). For our LLZTO-KF|Li interface, the forming LiF can prevent the further reaction between KF and Li at the interface and the forming K does not affect the insulation of the interlayer. Therefore, the ionic-conducting and electron-blocking layer consisted of LiF and KF was *in situ* generated after LLZTO-KF reacting with melted lithium metal.

Figure R10. Liquid metal-salt phase equilibria in the alkali metal-fluoride systems (LiF-Li and KF-K systems) ^[1].

Finally, the results, discussions and figures were modified in the revised Main Text in blue text according to this comment. (Page 5-6)

“Next, the reaction between KF layer and lithium metal was studied scientifically. For our solid-state lithium metal cells, the lithium anode was prepared by dipping the LLZTO-KF pellet into molten lithium, where lithium metal was tightly bonded to pellet and it was difficult to separate them apart to observe the phase evolution of the interlayer material. In order to observe the reaction product of Li and KF, we have designed two kinds of experiments. (1) KF powder react with melted Li metal (Sample 1#): put some KF powder into melted Li metal on stainless steel substrate with 300 °C hotplate for full reaction. (2) KF green pellet react with melted Li metal (Sample 2#): make a KF green pellet by a uniaxial pressure of 300 MPa, and put it into melted Li metal for 30 min. Then take out the KF pellet from melted Li (KF surface was covered by Li metal), and put on the 300 °C hotplate for full reaction. Then, XRD and grazing incidence XRD (GI-XRD) were carried out with the two samples. For Sample 1#, strong XRD peaks of Li (PDF#15-0401) and KF (PDF#85-1314) were collected as shown in Figure S7a (black line). More importantly, the weak XRD peaks of reaction product LiF (PDF#45-1460) were observed successfully.

For a clearer detection of the reaction products, GI-XRD was performed with an incidence angle of 1°. And stronger peaks of LiF were observed clearly (Figure S7b, black line). For Sample 2#, strong and clear peaks of Li, KF and LiF were detected from XRD (Figure S7a blue line) and GI-XRD patterns (Figure 2a). In addition, the phase structure of LLZTO-KF/Li sample (dipping LLZTO-KF pellet into melted lithium fleetly) was verified by synchrotron radiation GI-XRD with different incidence angle of 0.2°, 0.5° and 1° (Figure S8 and Figure 2b). Thanks to the high energy and high resolution of the synchrotron radiation X-ray, the phase structure of interlayer with very low content has been successfully detected. Figure 2b displayed the enlarged synchrotron radiation GI-XRD pattern at 1°, except the LLZTO substrate, the peaks of KF (28.9°, 33.5° and 48.1°), LiF (38.7° and 45°) and Li (36.2°) were observed clearly.

Moreover, to verify the reaction between the KF layer and Li clearly and easily, depth-profiling X-ray photoelectron spectroscopy (XPS) analysis was performed to obtain the composition information in different depth by ion etching. We firstly thermally evaporate ~100 nm lithium metal on the substrate, and then thermally evaporate a ~50 nm KF layer on the Li metal surface in glovebox, as shown in Figure 2c inset. Before etching, as shown in Figure 2c-e, classical K-F signal was clearly collected on the top layer (KF layer) surface of F 1s and K 2p regions, while no markedly Li signal was detected on the sample surface. With the etching depth increasing, the signal of generated LiF began to appear and gradually increased according to the XPS spectrum of F 1s and Li 1s. Moreover, the K-O signal was collected at 50 nm depth and it became stronger at 100 nm (Figure 2d), where the K-O signal come from the generated K metal (Notably, K is the active metal, always showing K-O signal in XPS result). For Li 1s regions, as displayed in Figure 2e, lithium metal signal began to appear when etching depth arriving Li metal layer (50 nm) and lithium metal peak occupied a higher proportion when arriving the interior lithium layer (100 nm). More intuitive evidence and discussion were given in Figure S9 and Note S1.

KF can react with molten Li metal at high temperatures according to Reaction 1:

According to the LiF-Li and KF-K phase diagrams as shown in Figure S10⁴¹, the forming LiF does not dissolve into Li and forming K does not dissolve into KF. For our LLZTO-KF/Li interface, during assembling LLZTO-KF with Li metal anode, KF reacted with molten Li metal partially, and the forming LiF could prevent the further reaction between KF and Li at the interface. And, the forming K with very low melting pointing might diffuse into molten Li metal during reaction, which did not affect the insulation of the interlayer. Therefore, the results of XRD, GI-XRD and depth-profiling XPS supported the above reaction mechanism. The in-situ reaction of KF and Li metal improves the lithiophilic ability and reduces

the interface impedance, resulting in excellent electrochemical performance.”

Figure 2 in revised Main Text. Reaction mechanism of Li and KF buffer layer. (a) The GI-XRD patterns with an incidence angle of 1° of Li-KF reaction product under Kapton film. (b) Synchrotron radiation GI-XRD patterns at the incidence angle of 1° . (c-e) XPS spectrum of F 1s (c), K 2p (d) and Li 1s (e) regions with various etching depth of 0, 20, 50 and 100 nm, respectively. The inset in (c) is the structure schematic diagram of Substrate|Li|KF samples.

Figure S7 in revised supplementary information. The XRD (a) and GI-XRD (b) patterns with an incidence angle of 1° of Li-KF reaction product under Kapton film.

Figure S8 in revised supplementary information. Synchrotron radiation GI-XRD patterns (a) and the enlarged picture (b) of LLZTO-KF|Li sample at the incidence angle of 0.2° , 0.5° and 1° .

Figure S9 in revised supplementary information. (a) The photos of LLZTO-KF|Li pellet with/without heat treatment. (b-f) XPS spectrum of Li 1s, F 1s, K 2p, La 3d and Zr 3d regions of LLZTO-KF|Li pellet with/without heat treatment, respectively.

Figure S10 in revised supplementary information. Liquid metal-salt phase equilibria in the alkali metal-fluoride systems (LiF-Li and KF-K systems). [1]

Question 6. The KF thin layer shows reduced electronic conductivity, increasing resistance and overpotential in Li symmetric cells. Why did KF-LLZTO show lower overpotential in the early cycle stages compared to LLZTO, despite the higher overpotentials observed under 0.2 mA cm^{-2} (0.1 mAh cm^{-2}) in both LLZTO and KF-LLZTO cells?

Response: Thanks for point it out. As we discussed above, the KF layer partly reacted with Li metal to form LiF and K. And the forming LiF prevented the further reaction between KF and Li at the interface, generating the ionic-conducting and electron-blocking layer consisted of LiF and KF. Therefore, the buffer layer can not only reduce electronic conductivity, but also conduct Li ion and improve lithium wettability, showing low interface resistance, high ionic conductivity, and low overpotential in Li symmetric cells. For higher overpotentials in both LLZTO and KF-LLZTO cells, the overpotential of LLZTO-KF cells is $\sim 48 \text{ mV}$ under 0.2 mA cm^{-2} , lower than that of 64 mV of LLZTO cells. In addition, the bare-LLZTO|Li interface will continue to deteriorate during the cycle process, while LLZTO-KF|Li cells can cycle stably.

Reference

[1] Dworkin, A. S., H. R. Bronstein, and M. A. Bredig. "Miscibility of metals with salts. vi. lithium-lithium halide systems1." *The Journal of Physical Chemistry* 66.3 (1962): 572-573.

REVIEWER COMMENTS

Reviewer #1 (Remarks to the Author):

I recommend to accept the paper in its present form without further revisions. The authors have well addressed all my comments.

Reviewer #2 (Remarks to the Author):

The comments in my previous review have been well addressed. I agree with the publication of this revised manuscript in Nature Communications.

Reviewer #3 (Remarks to the Author):

The authors address all issues that the reviewer raises, so I recommend that it be published in Nature Communications. However, I have a few more questions in simulations, which the authors added and modified.

Q1) In DFT calculations, the composite structure of LLZTO/Li, KF/Li, and LiF/Li were generated for atomic simulation. However, the authors did not describe the simulation details, such as crystal structures and size of composites. Due to the limit of the number of atoms, I think the DFT calculations were performed with single K-point sampling. Although thermodynamic calculations, such as interfacial formation energy, are not strongly affected by lower K-point sampling, electronic convergence is needed to check for DOS and electrostatic potential carefully.

Q2) In the DOS analysis, the authors said that electrical conduction was validated by the electronic population. The PDOSs of Li in Li metal (black line) show similar metal behavior in 3 cases. The KF and LF also show conductive behavior, and the LLZTO shows an insulator characteristic. I can not understand the explanation: "The electronically insulating nature is further confirmed by DOS results for KF|Li and LiF|Li, shown in Figure 4g-h."

Q3) Figure 4a-c show the charge density difference between solid electrolyte and Li metal. The authors did not explain the details of Figure 4a-c. The meaning of isosurface and isovalue is needed.

Q4) In the first-round revision, referee 1 and I addressed the COMSOL simulation issues. The COMSOL modeling in LIB modules defines the geometries as electrolytes and electrodes. Each phase of the electrolyte and electrode shows the different current densities of electrolyte and electric, respectively. However, this simulation shows current density, not electrolyte current density or electrical current density. I'm not sure that the simulation results suggest a KF layer effect considering interfacial physics, not a geometrical effect using only electric simulation. Authors must list the material properties and equations used in this study.

I think all questions related to the experiments and analysis are clear. I hope that the authors can address the simulation and modeling issues.

Title: Extreme electron-blocking interface for garnet-based solid-state lithium-metal batteries with superior long lifespan

Corresponding author: Wei Liu

Manuscript ID: NCOMMS-23-48483A

Response to the reviewers' comments

First of all we would like to thank the reviewers for the time and effort in reviewing this manuscript. The comments are all valuable and constructive for revising and improving our paper, as well as the important guiding significance to our researches. We have provided the point-by-point responses below in blue text while keeping the reviewer's comments in black.

REVIEWER COMMENTS

Reviewer #1 (Remarks to the Author):

I recommend to accept the paper in its present form without further revisions. The authors have well addressed all my comments.

Reviewer #2 (Remarks to the Author):

The comments in my previous review have been well addressed. I agree with the publication of this revised manuscript in Nature Communications.

Reviewer #3 (Remarks to the Author):

The authors address all issues that the reviewer raises, so I recommend that it be published in Nature Communications. However, I have a few more questions in simulations, which the authors added and modified.

Question 1: In DFT calculations, the composite structure of LLZTO/Li, KF/Li, and LiF/Li were generated for atomic simulation. However, the authors did not describe the simulation details, such as crystal structures and size of composites. Due to the limit of the number of atoms, I think the DFT calculations were performed with single K-point sampling. Although thermodynamic calculations, such as interfacial formation energy, are not strongly affected by lower K-point sampling, electronic convergence is needed to check for DOS and electrostatic potential carefully.

Response: We thank the reviewer's comments and constructive suggestions. We have listed a table (Table S6) to describe the computational details and checked the electronic convergence for DOS and electrostatic potential carefully.

Finally, the description was modified in the **revised Main Text** in blue text (Page 13), and the computational details was shown in Table S6 in the **revised supplementary information** in blue text according to this comment.

“DFT calculations. ...The structural and electronic properties were calculated in the LLZTO bulk, LLZTO slab, KF slab and LLZTO/KF interface. The plane-wave cutoff energy of 400 eV and uniform G-centered k-points meshes with a resolution of $2\pi \times 0.04 \text{ \AA}^{-1}$ was employed. The computational details were listed in Table S6 in the supplementary information. ...”

Table S6 in revised supplementary information. The DFT calculation details for LLZTO/Li, KF/Li, and LiF/Li interface.

Parameters	Settings		
Pseudopotential type	Projector augmented wave (PAW)		
Exchange-correlation functional	Generalized gradient approximation (GGA) of Perdew–Burke–Ernzerh (PBE)		
Valence electron configurations	Li-2s¹, O-2s²2p⁴, F-2s²2p⁵, K-4s¹, Zr-5s²4d², La-5d¹6s², Ta-5d³6s²		
Precision mode	Accurate		
Cutoff energy (eV)	400		
Electronic convergence (eV)	1×10⁻⁴		
Systems	LLZTO/Li	KF/Li	LiF/Li
Number of atoms	152 Li 24 La 12 Zr 4 Ta 96 O 288 in total	72 K 72 F 96 Li 240 in total	72 F 168 Li 240 in total
k-point meshes for relaxation	2×2×1	2×2×1	2×2×1
k-point meshes for DOS calculation	4×4×2	6×6×3	6×6×3
Center of the meshes	Gamma centered mesh.		

Question 2: In the DOS analysis, the authors said that electrical conduction was validated by the electronic population. The PDOSs of Li in Li metal (black line) show similar metal behavior in 3 cases. The KF and LF also show conductive behavior, and the LLZTO shows an insulator characteristic. I can not understand the explanation: “The electronically insulating nature is further confirmed by DOS results for KF|Li and LiF|Li, shown in Figure 4g-h.”

Response: We thank the reviewer's comments. We are sorry for this confusion, since we should have presented the scale ranges for the DOS axis in Figure 4g-i. In the first-round revision, the vertical scale ranges were from 0 to 10 for Figure 4g (KF|Li) and 4h (LiF|Li), whereas it was from 0 to 50 for Figure 4i (LLZTO|Li). Accordingly, the insulating characteristic cannot be compared directly among the three DOS figures. To avoid misleading, we employed the same vertical scale range (0 to 100) and redrew the three DOS figures. Please see the figures below. The **red lines** indicated the insulator characteristic for KF and LiF.

Figure 4 in revised Main Text. (g-i) The corresponding density of states (DOS) of interface.

In addition, we zoomed in the DOS under a narrow range of 0 to 10. Please see the figures below. The DOS data (**red lines**) across Fermi level are 0.33825, 0.48096, and 0.56837 for KF, LiF and LLZTO, respectively, confirming a better insulating nature for KF and LiF, as compared to LLZTO.

Figure S15 in revised supplementary information. Density of states for the interfaces of KF|Li (a), LiF|Li (b) and LLZTO|Li (c).

Finally, the discussions and figures were modified in blue text in revised Main Text according to this comment (Page 8 and 21). And the enlarged figures have been presented as Figure S15 in revised supplementary information according to this comment.

“The electronically insulating nature is further confirmed by DOS results for KF|Li and LiF|Li, shown in Figure 4g-h and the enlarged figures in Figure S15a-b in supplementary information. In contrast, there is no barrier ($\Delta E=8.92$ eV>0, Figure 4f) to the transfer of electrons from the Li metal to the bare LLZTO electrolyte. In the case of bare LLZTO|Li interface, electrons and Li atoms preferentially deposit within the LLZTO, and this behavior also corroborated by DOS results (Figure 4i and figure S15c).”

Question 3: Figure 4a-c show the charge density difference between solid electrolyte and Li metal. The authors did not explain the details of Figure 4a-c. The meaning of isosurface and isovalue is needed.

Response: We thank the reviewer’s constructive suggestions. The meaning of isosurface and isovalue is added the figure caption as following (Page 21):

Figure 4 in revised Main Text. Electron-blocking property of the interface. (a-c) The structure and charge transfer of KF|Li interface (a) LiF|Li interface (b) and bare LLZTO|Li interface (c). The yellow bubbles indicate electron accumulation, and blue bubbles represent electron depletion, the isosurface value is $0.005 e \text{ \AA}^{-3}$. (d-f) The corresponding electrostatic potential profiles of interface. (g-i) The corresponding density of states (DOS) of interface.

Question 4: In the first-round revision, referee 1 and I addressed the COMSOL simulation issues. The COMSOL modeling in LIB modules defines the geometries as electrolytes and electrodes. Each phase of the electrolyte and electrode shows the different current densities of electrolyte and electric, respectively. However, this simulation shows current density, not electrolyte current density or electrical current density. I'm not sure that the simulation results suggest a KF layer effect considering interfacial physics, not a geometrical effect using only electric simulation. Authors must list the material properties and equations used in this study.

I think all questions related to the experiments and analysis are clear. I hope that the authors can address the simulation and modeling issues.

Response: Thanks for point it out. In fact, we obtained the electrode current densities figures and electrolyte current densities figures from COMSOL simulation as shown in Figure R1. In the revised Main Text, we merged the two figures into one current density distribution figure for brief statement.

Figure R1. The finite element analysis results of electrolyte current density distribution (a₁ and b₁) and electrode current density distribution (b₁ and b₂) in LLZO-KF|Li interface (a) and LLZO|Li interface (b).

Moreover, the difference between the two models is the interlayer. For LLZTO-KF|Li interface, the *in-situ* formed LiF/KF interlayer could restrain the lithium dendrite formation, improve the Li metal wettability, and homogenize the ion transport, which bring the homogeneous current density distribution. Therefore, the whole interface is the “Internal electrode surface”, where homogeneous Li plating/stripping process was observed. While, at LLZTO|Li interface with poor interface contact, heterogeneous Li

plating/stripping process and hotspots occurred. Therefore, there may be some inactive “Internal electrode surface”, where Li plating/stripping process was restricted. And the simulation details are given in Table S5 as below:

Table S5 in revised supplementary information. The COMSOL simulation details.

Parameters	Value
Geometrical size	4×3 μm ²
Electrolyte conductivity	5×10 ⁻⁴ S cm ⁻¹
Electrode conductivity	1×10 ⁷ S cm ⁻¹
Applied current density	0.1 mA cm ⁻²
Reaction electron number	1
Reference exchange current density (LLZTO Li)	100 A m ²
Reference exchange current density (LLZTO-KF Li)	1000 A m ²

The simulation results were obtained from COMSOL software with “Lithium-Ion Battery interface”. The charge balance model was “Single ion conductor”. Electrochemical reaction kinetics at the electrode-electrolyte interface could be described by Butler–Volmer equation:

$$i_{loc} = i_0 \left(\exp\left(\frac{\alpha_a F \eta}{RT}\right) - \exp\left(\frac{-\alpha_c F \eta}{RT}\right) \right)$$

where i_{loc} is the actual exchange current density, i_0 represents exchange current density, α_a is the anodic charge transfer coefficient and α_c is the cathodic charge transfer coefficient, η is the overpotential, T is the system temperature, F is the ideal gas constant.

Finally, the descriptions and figures were modified in the revised Main Text and revised supplementary information in blue text according to this comment. (Page 13)

“COMSOL somulation. The simulation results were obtained from COMSOL software

with “Lithium-Ion Battery interface”. The charge balance model was “Single ion conductor”. A simplified cell model was used to simulate the electric field distribution at the interface between Li anode and LLZTO solid state electrolyte with COMSOL Multiphysics software. In the $4 \times 3 \mu\text{m}^2$ 2D geometry models (Figure S27), the top area was LLZTO with or without electron-blocking and lithiophilic layer, the bottom area was the Li anode. The electron blocking and lithiophilic layer could improve the interface reaction uniformly. Meanwhile, the control model without buffer layer cause heterogeneous reaction at the interface. In the simulated deposition process, the ionic conductivity is $5 \times 10^{-4} \text{ S cm}^{-1}$, the applied average current density is 0.1 mA cm^{-2} , and the colour bar denotes the current density. Electrochemical reaction kinetics at the electrode-electrolyte interface could be described by Butler–Volmer equation:

$$i_{loc} = i_0 \left(\exp\left(\frac{\alpha_a F \eta}{RT}\right) - \exp\left(\frac{-\alpha_c F \eta}{RT}\right) \right)$$

Where i_{loc} is the actual exchange current density, i_0 represents exchange current density, α_a is the anodic charge transfer coefficient and α_c is the cathodic charge transfer coefficient, η is the overpotential, T is the system temperature, F is the ideal gas constant.”

Table S5. The COMSOL simulation details.

Parameters	Value
Geometrical size	$4 \times 3 \mu\text{m}^2$
Electrolyte conductivity	$5 \times 10^{-4} \text{ S cm}^{-1}$
Electrode conductivity	$1 \times 10^7 \text{ S cm}^{-1}$
Applied current density	0.1 mA cm^{-2}
Reaction electron number	1
Reference exchange current density (LLZTO)	100 A m^2
Reference exchange current density (LLZTO-KF)	1000 A m^2

REVIEWERS' COMMENTS

Reviewer #3 (Remarks to the Author):

I also recommend to accept the paper in its present form without further revisions. The authors have well addressed all my comments and simulation details.